# Suppressing neutrophil itaconate production attenuates *Mycoplasma pneumoniae* pneumonia

Cui Wang[1,2]☯, Jun Wen[3]☯, Zijun Yan[1,2], Yujun Zhou[1,2], Zhande Gong[1,2], Ying Luo[1,2], Zhenkui Li[1,2], Kang Zheng[4], Haijun Zhang[5], Nan Ding[1,2], Chuan Wang[1,2], Cuiming Zhu[1,2], Yimou Wu[1,2], Aihua Lei ◉[1,2]*

1 Institute of Pathogenic Biology, School of Basic Medical Sciences, Hengyang Medical School, University of South China, Hengyang, Hunan, China, 2 Hunan Provincial Key Laboratory for Special Pathogens Prevention and Control, University of South China, Hengyang, Hunan, China, 3 Department of pediatrics, The First Affiliated Hospital, Hengyang Medical School, University of South China, Hengyang, China, 4 Affiliated Hengyang Hospital of Hunan Normal University & Hengyang Central Hospital, Hengyang, Hunan, China, 5 Department of Cardiology, The First Affiliated Hospital, Hengyang Medical School, University of South China, Hengyang, China

☯ These authors contributed equally to this work.
* leiaihua18@usc.edu.cn

**Data Availability Statement:** The authors confirm that all data underlying the findings are fully available without restriction. All relevant data are

## Abstract

*Mycoplasma pneumoniae* is a common cause of community-acquired pneumonia in which neutrophils play a critical role. Immune-responsive gene 1 (IRG1), responsible for itaconate production, has emerged as an important regulator of inflammation and infection, but its role during *M. pneumoniae* infection remains unknown. Here, we reveal that itaconate is an endogenous pro-inflammatory metabolite during *M. pneumoniae* infection. *Irg1* knockout (KO) mice had lower levels of bacterial burden, lactate dehydrogenase (LDH), and pro-inflammatory cytokines compared with wild-type (WT) controls after *M. pneumoniae* infection. Neutrophils were the major cells producing itaconate during *M. pneumoniae* infection in mice. Neutrophil counts were positively correlated with itaconate concentrations in bronchoalveolar lavage fluid (BALF) of patients with severe *M. pneumoniae* pneumonia. Adoptive transfer of *Irg1* KO neutrophils, or administration of β-glucan (an inhibitor of *Irg1* expression), significantly attenuated *M. pneumoniae* pneumonia in mice. Mechanistically, itaconate impaired neutrophil bacterial killing and suppressed neutrophil apoptosis via inhibiting mitochondrial ROS. Moreover, *M. pneumoniae* induced *Irg1* expression by activating NF-κB and STAT1 pathways involving TLR2. Our data thus identify *Irg1*/itaconate pathway as a potential therapeutic target for the treatment of *M. pneumoniae* pneumonia.

## Author summary

*M. pneumoniae* is a common human pathogen that causes community-acquired pneumonia. Neutrophil infiltration has been widely recognized as a characteristic of *M. pneumoniae* pneumonia; however, the interaction between neutrophils and *M. pneumoniae* is not

within the paper and its Supporting Information files.

**Funding:** This study was supported by National Natural Science Foundation of China (no. 82371790 to AL), Natural Science Foundation of Hunan Province (no. 2022JJ20034 to AL; no. 2022JJ50165 to HZ), Research Foundation of Education Bureau of Hunan Province (no. 22B0412 to AL), and Research Foundation of Health Commission of Hunan Province (no. 202202074638 to AL). The funders had no role in study design, data collection and analysis, decision to publish, or preparation of the manuscript.

**Competing interests:** The authors have declared that no competing interests exist.

completely understood. Here we demonstrate a critical role of the *Irg1*/itaconate metabolic pathway in neutrophils in the pathogenesis of *M. pneumoniae* pneumonia. We observe a positive correlation between neutrophil counts and itaconate levels in BALF from patients with severe *M. pneumoniae* pneumonia. We find that *M. pneumoniae* infection induces itaconate production in neutrophils, which dampens neutrophil bactericidal activity and impedes neutrophil apoptosis through inhibiting mitochondrial ROS (mtROS). Suppressing neutrophil itaconate production mitigates *M. pneumoniae* pneumonia in mice. We also find that *M. pneumoniae* induces *Irg1* expression in neutrophils by activating NF-κB and STAT1 signaling pathways. Therefore, our findings suggest that targeting neutrophil itaconate production may be an effective therapeutic strategy for the treatment of *M. pneumoniae* pneumonia in humans.

## Introduction

*Mycoplasma pneumoniae* is an atypical bacterium that causes lung infection and community-acquired pneumonia, especially in children. *M. pneumoniae* pneumonia outbreaks are a serious public health issue due to the alarming rise of antibiotic-resistant strains worldwide [1,2]. Novel therapies are urgently needed to deal with antibiotic resistance in mycoplasmas. While immune responses play an important role in eliminating *M. pneumoniae* and mediating lung inflammation and injury, the interaction mechanism between *M. pneumoniae* infection and host immune response remains largely unknown.

Neutrophils are the major phagocytes that act as the first line of innate immune defense against respiratory infections [3,4]. It is widely recognized that neutrophils kill bacteria mainly through phagocytosis, releasing neutrophil extracellular traps (NETs), producing reactive oxygen species (ROS), and degranulation [5]. Abundant neutrophil infiltration is a typical feature at the early stage of *M. pneumoniae* pneumonia [6–10]. However, recent studies have shown that neutrophils are key players in promoting *M. pneumoniae* pneumonia instead of effectively eliminating *M. pneumoniae* [11,12]. The severity of *M. pneumoniae* pneumonia in patients is closely associated with the increased number of neutrophils in bronchoalveolar lavage fluid (BALF) [6,7,10]. The number of neutrophils may be useful in the prognosis of *M. pneumoniae* pneumonia [10,13]. Moreover, *M. pneumoniae* infection can induce neutrophils to produce proinflammatory cytokines, such as IL-1β, and TNF-α [11,14]. Notably, studies have found that *M. pneumoniae* can evade neutrophil clearance via avoiding neutrophil phagocytosis, and degrading NETs by secreting the nuclease Mpn491 [15,16]. Although targeting neutrophils may improve *M. pneumoniae* pneumonia [12], the precise role of neutrophils during *M. pneumoniae* infection remains unclear.

Metabolic reprogramming is critical for myeloid cell functions [17,18]. Emerging evidence has demonstrated that endogenous metabolites in myeloid cells hold crucial immunoregulatory actions during bacterial infections [19]. One of the most important metabolites receiving much attention in recent years is itaconate, which is produced by the decarboxylation of cis-aconitate through immune-response gene 1 (IRG1; also known as aconitate decarboxylase 1, ACOD1) in the tricarboxylic acid cycle under inflammatory conditions [20,21]. The expression of *Irg1* in myeloid cells can be induced by a variety of factors, such as pathogen infections and Toll-like receptor (TLR) agonists [20]. During inflammation and infection, *Irg1*/Itaconate pathway has been shown to control myeloid cell function through multiple mechanisms, such as succinate dehydrogenase (SDH) inhibition, nuclear factor erythroid 2-like 2 (NFE2L2 or NRF2) activation, and modulation of oxidative stress [20,22]. Of note, *Irg1*/Itaconate pathway

displays a dual role in infection and inflammation [20,23]. On the one hand, itaconate production confers a protective role by limiting pathogen infections and inflammation [24,25]. For example, during *Mycobacterium tuberculosis* (*Mtb*) infection, *Irg1*-deficient mice exhibits a higher bacterial burden and more severe lung disease [25]. On the other hand, abnormal IRG1 expression can also exert a deleterious effect through promoting inflammation or mediating immunosuppression under pathological conditions [26–30]. For instance, inhibition of IRG1 expression reduces lung inflammation and injury in respiratory syncytial virus-infected mice [26]. Moreover, targeting IRG1 can boost antitumor immunity by reversing the immunosuppressive function of tumor-infiltrating myeloid cells [30,31]. Recently, Tomlinson et al., showed that activation of *Irg1*/itaconate pathway inhibits neutrophil oxidative burst and glycolysis during *Staphylococcus aureus* infection [27]. In addition, itaconate-producing neutrophils are found to contribute to local and systemic inflammation following trauma [29]. These previous reports suggest that neutrophil itaconate production plays a critical role in regulating inflammation and innate immunity. However, whether *Irg1*/Itaconate pathway controls the severity of *M. pneumoniae* pneumonia is unknown.

In this study, we demonstrate that activation of *Irg1*/itaconate pathway is critical for promoting *M. pneumoniae* pneumonia. Neutrophils are the major cells producing itaconate during *M. pneumoniae* infection. Adoptive transfer of *Irg1* KO neutrophils significantly alleviates *M. pneumoniae* pneumonia in mice. β-glucan treatment can reduce *Irg1* expression in neutrophils and attenuate *M. pneumoniae*-induced lung inflammation. Importantly, *M. pneumoniae* induces neutrophil itaconate production that impairs bactericidal activity and impedes neutrophil apoptosis via inhibiting mitochondrial ROS (mtROS). Mechanically, *M. pneumoniae* induces *Irg1* expression by MyD88/NF-κB and STAT1 signaling pathways involving TLR2. Thus, our findings identify *Irg1*/itaconate pathway as a potential target for the treatment of *M. pneumoniae* pneumonia.

## Results

### Itaconate production contributes to *M. pneumoniae* pneumonia in mice

To explore the role of IRG1/itaconate metabolic pathway during *M. pneumoniae* infection, we intranasally challenged BALB/c mice with *M. pneumoniae* as described previously [9,32]. Compared with mock infection, the mRNA expression of *Irg1* gene in lung tissue was significantly increased at day 1 postinfection (p.i.) and day 3 p.i., returning to almost baseline levels at day 7 p.i. (Fig 1A). Meanwhile, the IRG1 protein expression in lung tissue by western blot displayed a similar change (Fig 1B). Consistent with IRG1 expression, the itaconate levels in BALF were dramatically elevated at day 1, especially at day 3 p.i. and returned to baseline levels at day 7 p.i. (Fig 1C). These results indicate that *M. pneumoniae* infection upregulates *Irg1*/Itaconate pathway in mice.

To investigate whether itaconate production is involved in *M. pneumoniae* pneumonia, we compared the response of wild-type (WT) and *Irg1* knockout (KO) C57BL/6J mice to *M. pneumoniae* infection combined with intranasal administration with itaconate or PBS for three consecutive days and mice were sacrificed at day 3 p.i. As expected, *M. pneumoniae* infection caused a significant increase in itaconate levels in the BALF of WT mice but not *Irg1* KO mice, whereas itaconate administration restored itaconate levels in infected KO mice (Fig 1D). Notably, we found that *Irg1* KO mice had a lower load of *M. pneumoniae* bacteria in BALF and lung tissue than that from WT mice after *M. pneumoniae* infection, whereas exogenously added itaconate almost restored *M. pneumoniae* load in *Irg1* KO mice (Fig 1E and 1F). Meanwhile, after *M. pneumoniae* challenge, *Irg1* KO mice showed lower levels of proinflammatory cytokines including IL-1β, IL-6, and TNF-α (Fig 1G), lactate dehydrogenase (LDH) (Fig 1H),

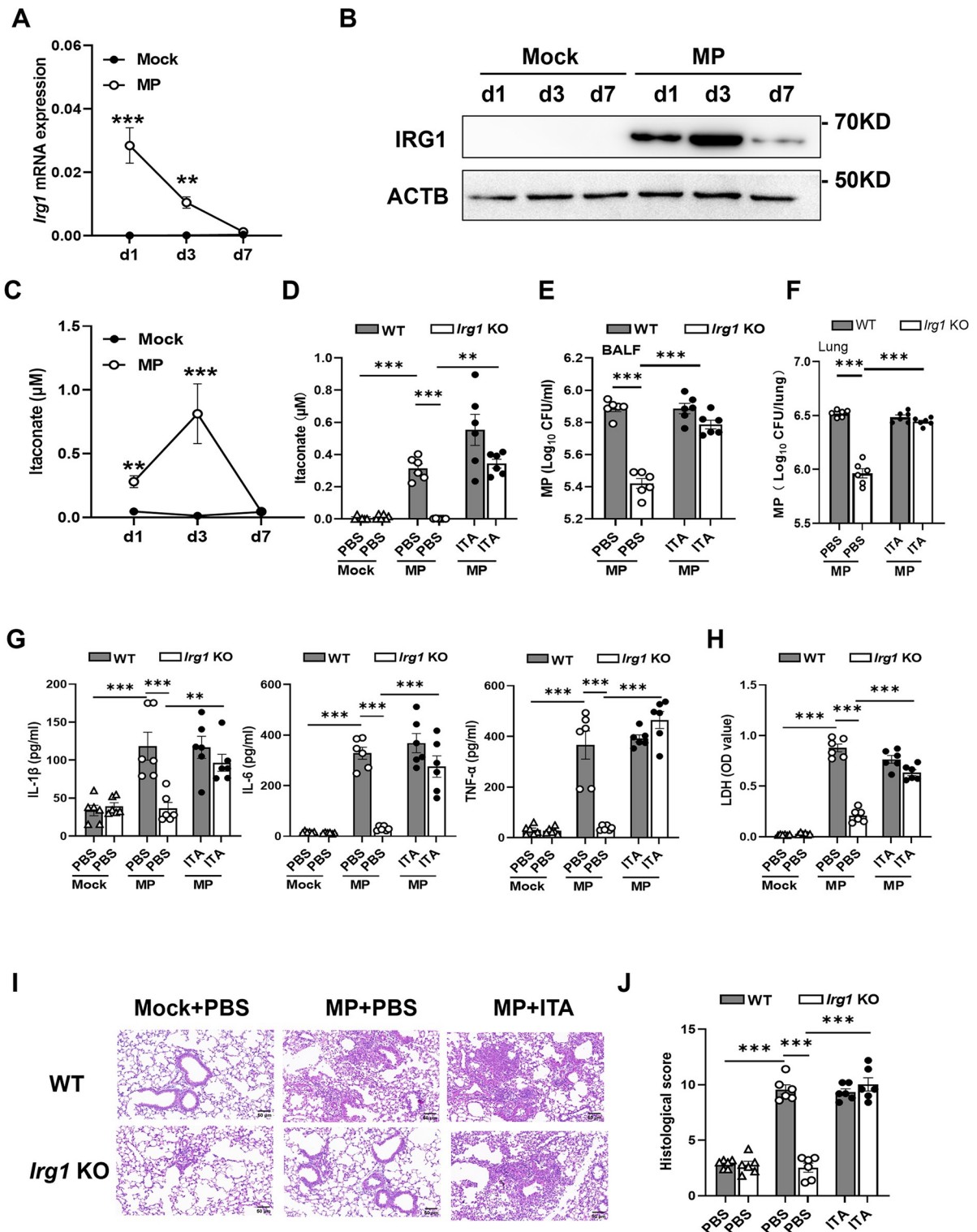

**Fig 1. The *Irg1*/Itaconate pathway is upregulated during *M. pneumoniae* infection, and deficiency of *Irg1* alleviates *M. pneumoniae* pneumonia in mice.** (A) Dynamic changes in *Irg1* mRNA expression during *M. pneumoniae* (MP) infection in BALB/c mice (n = 4 for each mock group and n = 8 for each infection group), pooled from three independent experiments. Data are presented as mean ± SEM. Statistical significance tested by unpaired, two-tailed Student's *t* test. (B) IRG1 protein expression in the mouse lung tissue during infection, representative of three experiments. (C) Dynamic changes in itaconate production in the BALF during infection (n = 4 for each mock group

and n = 7 for each infection group), pooled from three independent experiments. Data are presented as mean ± SEM. Statistical significance tested by unpaired, two-tailed Student's *t* test. (D-J) C57 WT and *Irg1* KO mice were intranasally infected with *M. pneumoniae*, and 2 h later were intranasally administrated with itaconate (ITA) (10 mg/kg/day) or PBS for 3 consecutive days. Mice were sacrificed on day 3 (n = 6 per group). Data were pooled from two independent experiments. (D) Itaconate concentrations in BALF. (E) *M. pneumoniae* colony-forming units (CFU) in BALF. (F) *M. pneumoniae* CFU in lung homogenates. (G) The amount of IL-1β, IL-6 and TNF-α in BALF by ELISA. (H) BALF LDH levels. (I) Representative H&E staining of lung tissue. Scale bar, 50 μm. (J) Histological scores. Data are presented as mean ± SEM. Statistical significance tested by one-way ANOVA test (*, p<0.05; **, p<0.01; ***, p<0.001).

and an attenuated lung inflammation as evidenced by H&E staining of lung tissues compared to WT controls (Fig 1I and 1J). These parameters in infected *Irg1* KO mice were also restored after exogenous itaconate treatment (Fig 1E–1J). Consistent with previous reports [12,33], *M. pneumoniae* infection resulted in an accumulation of large numbers of neutrophils in the BALF and the lung tissue in mice when compared with mock infection (S1A–S1C Fig). Interestingly, the number of neutrophils in BALF and lung from *Irg1* KO mice was much lower than that from WT mice after *M. pneumoniae* infection, while exogenously added itaconate restored the neutrophil numbers in *Irg1* KO mice (S1A–S1C Fig). Meanwhile, the number of alveolar macrophages in BALF and lung tissue was not altered between WT and *Irg1* KO mice after *M. pneumoniae* challenge without or with exogenous itaconate treatment (S1D and S1E Fig). Taken together, these results showed that itaconate production contributes to *M. pneumoniae* pneumonia in mice.

## IRG1 is primarily expressed by neutrophils during *M. pneumoniae* infection

As studies have demonstrated that myeloid cells express IRG1 during pathogen infection [27], we first determined the cell count of neutrophils, monocytes, and alveolar macrophages during *M. pneumoniae* lung infection in mice by flow cytometry. Notably, we found that the number of neutrophils was dramatically increased at day 1 and day 3 p.i., and returned to baseline levels at day 7 p.i. in BALB/c mice (Fig 2A), similar to that of *Irg1* mRNA expression in the lung tissue. Meanwhile, the number of lung neutrophils far outweighed the number of monocytes and alveolar macrophages at day 1 and day 3 p.i. (Fig 2A), suggesting that neutrophils may be the major source of itaconate during *M. pneumoniae* lung infection. Indeed, the lung neutrophils from *M. pneumoniae*-infected mice at day 1 and day 3, but not day 7, expressed a high level of *Irg1* mRNA expression, which was not expressed in the lung neutrophils from mock-infected mice (Fig 2B and 2C). Moreover, depletion of neutrophils from lung leukocytes derived from *M. pneumoniae*-infected mice on day 1 and day 3 almost removed the IRG1 protein (Fig 2D), though alveolar macrophages also expressed IRG1 protein (S2A Fig). For further confirmation, we depleted neutrophils by using anti-Ly6G antibody at day 1 p.i. in C57BL/6J mice and then measured *Irg1* expression in the lung tissue and itaconate production in BALF at day 3 p.i. Similar to previous studies [8,12], anti-Ly6G treatment significantly decreased the number of lung neutrophils compared with isotype IgG treatment in mice (S2B Fig), and did not affect *M. pneumoniae* burden in BALF (S2C Fig). Meanwhile, anti-Ly6G administration had no effect on the number of alveolar macrophages (S2D Fig) but caused more recruited monocytes compared to IgG isotype controls (S2E Fig). Despite the increase in monocytes, depletion of neutrophils significantly decreased the expression levels of *Irg1* mRNA (Fig 2E) and IRG1 protein (Fig 2F) in the mouse lung tissue, leading to a remarkably decline of the itaconate production in BALF (Fig 2G) from *M. pneumoniae*-infected mice.

In addition, we examined *Irg1* expression in mouse neutrophils at different time points after *M. pneumoniae* infection *in vitro*. We found neutrophils did not express *Irg1* within 4 h but expressed a high level of *Irg1* mRNA and IRG1 protein at 6 h and 12 h p.i. (S2F and S2G Fig). As

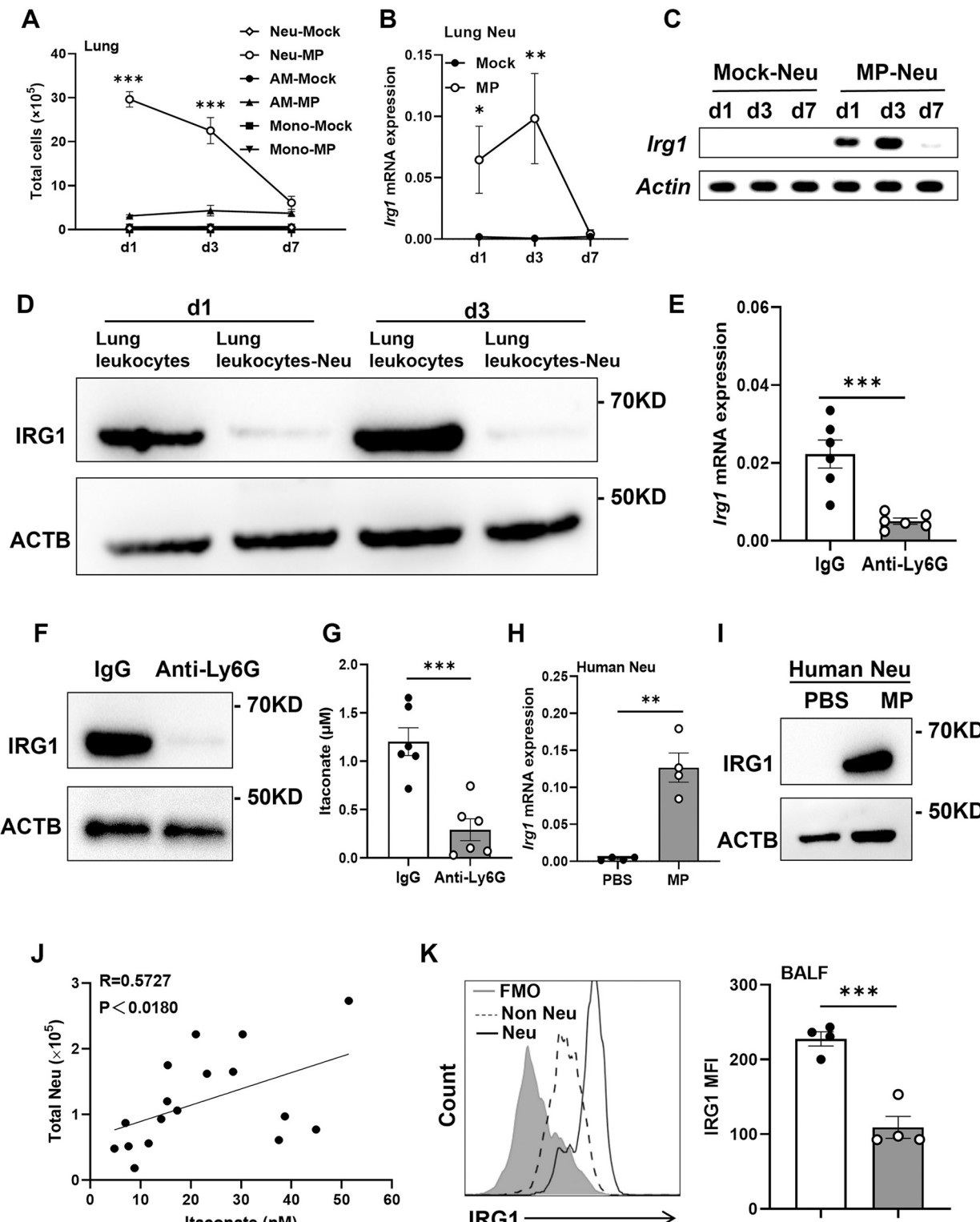

**Fig 2. Neutrophils are the main cells expressing *Irg1* and producing itaconate during *M. pneumoniae* infection.** (A) The number of CD45+CD11b+Ly6G+ neutrophils (Neu), CD45+CD11c+Siglec-F+ alveolar macrophages (AM) and CD45+CD11b+Ly6G-/lowLy6Chigh monocytes (Mono) in the mouse lung during *M. pneumoniae* (MP) infection (n = 4 to 8 per group). (B) The *Irg1* mRNA expression in the sorted mouse lung neutrophils during *M. pneumoniae* infection (n = 3 per group). (C) *Irg1* induction was determined by RT-PCR, representative of three experiments. PCR product length for *Irg1*: 189bp; PCR product length for *Actin*: 134bp. (D) IRG1 protein expression in the lung leukocytes and

lung leukocytes depleted of neutrophils (Lung leukocytes-Neu) from *M. pneumoniae*-infected mice at d1 and d3 by Western blot. Neutrophils were deleted from the lung leukocytes by magnetic-activated cell sorting. The experiment was performed three times independently. (E-G) Mice were intraperitoneally injected with anti-Ly6G antibodies or IgG control (0.2 mg/mouse) at d1 after *M. pneumoniae* infection and were sacrificed at d3 post infection (n = 6 per group). Data were pooled from two independent experiments. (E) *Irg1* mRNA expression in the mouse lung tissue. (F) The expression of IRG1 protein in lung tissue. (G) Itaconate concentrations in BALF. (H and I) *Irg1* mRNA and IRG1 protein expression in human neutrophils after 12 h of infection by *M. pneumoniae*, respectively, representative of three experiments. (J) Correlation analysis of BALF itaconate concentrations and the number of neutrophils in patients with severe *M. pneumoniae* pneumonia (n = 17). The correlative analysis was conducted using Spearman correlation. (K) Flow cytometric analysis of IRG1 expression in the CD45+CD15+CD16+ neutrophils and CD45+CD15-CD16- cells (Non-Neu) in the BALF from patients with severe *M. pneumoniae* pneumonia (n = 4). MFI: mean fluorescence intensity. Data are presented as mean ± SEM. Statistical significance tested by unpaired, two-tailed Student's *t* test (*, p<0.05; **, p<0.01; ***, p<0.001).

expected, *M. pneumoniae* infection induced neutrophils producing high levels of itaconate at 20 h p.i. (S2H Fig). Moreover, similar to live *M. pneumoniae*, heat-killed *M. pneumoniae* and *M. pneumoniae* lipid-associated membrane proteins (LAMP) can also induce mouse neutrophils to express IRG1 (S2I Fig), suggesting that *M. pneumoniae* directly induces *Irg1* expression in lung neutrophils in mice. Similarly, *M. pneumoniae* stimulated human neutrophils expressing *Irg1* mRNA and IRG1 protein (Fig 2H and 2I), as well as producing itaconate (S2J Fig). More importantly, we observed that itaconate concentrations in the BALF of patients with severe *M. pneumoniae* pneumonia were correlated positively with the number of BALF neutrophils (Fig 2J). Furthermore, flow analysis showed that BALF neutrophils had a higher level of IRG1 expression than non-neutrophils in BALF from severe patients (Fig 2K). Taken together, these data suggest that neutrophils are the predominant source of itaconate during *M. pneumoniae* lung infection.

## Adoptive transfer of *Irg1* KO neutrophils alleviates *M. pneumoniae* pneumonia in mice

To determine the role of itaconate-producing neutrophils during *M. pneumoniae* infection, naïve WT mice received bone marrow neutrophils from WT or *Irg1* KO mice and were challenged with *M. pneumoniae*. We confirmed the presence of a high proportion of transferred neutrophils (more than 70%) in BALF and lung by using CD45.1/2 mice as recipient mice (S3 Fig). After 3 days p.i., the number of neutrophils in BALF and lung tissue (Fig 3A and 3B), as well as itaconate levels (Fig 3C) was significantly reduced in the *M. pneumoniae*-infected WT recipient mice receiving *Irg1* KO neutrophils compared with those receiving WT neutrophils. We did not observe any difference in the weight loss between mice receiving WT neutrophils and mice receiving *Irg1* KO neutrophils after *M. pneumoniae* infection, although infection caused a slight decrease in mouse weight (S4 Fig). Of note, adoptive transfer of *Irg1* KO neutrophils, but not WT neutrophils, remarkably attenuated *M. pneumoniae* pneumonia in WT mice, as evidenced by significantly reduced *M. pneumoniae* burden (Fig 3D), proinflammatory cytokines (IL-1β, IL-6, and TNF-α) (Fig 3E), LDH levels (Fig 3F), and alleviated lung inflammation (Fig 3G and 3H). For further confirmation, we transferred bone marrow neutrophils from WT or *Irg1* KO mice into *Irg1* KO mice followed by *M. pneumoniae* infection. As expected, the *Irg1* KO mice receiving WT neutrophils had higher neutrophils numbers in BALF and lung tissue (S5A and S5B Fig), itaconate levels in BALF (S5C Fig), *M. pneumoniae* burden (S5D Fig), and LDH content (S5E Fig), potentiating to lung inflammation (S5F and S5G Fig) after 3 days of *M. pneumoniae* infection. Together, these data suggest that itaconate-producing neutrophils contribute to *M. pneumoniae* pneumonia in mice.

## IRG1 inhibitor β-glucan attenuates *M. pneumoniae* pneumonia in mice

As β-glucan, a fungal cell wall component, can serve as an inhibitor of *Irg1* expression [28], we next asked whether β-glucan could inhibit *Irg1* expression in neutrophils during *M.*

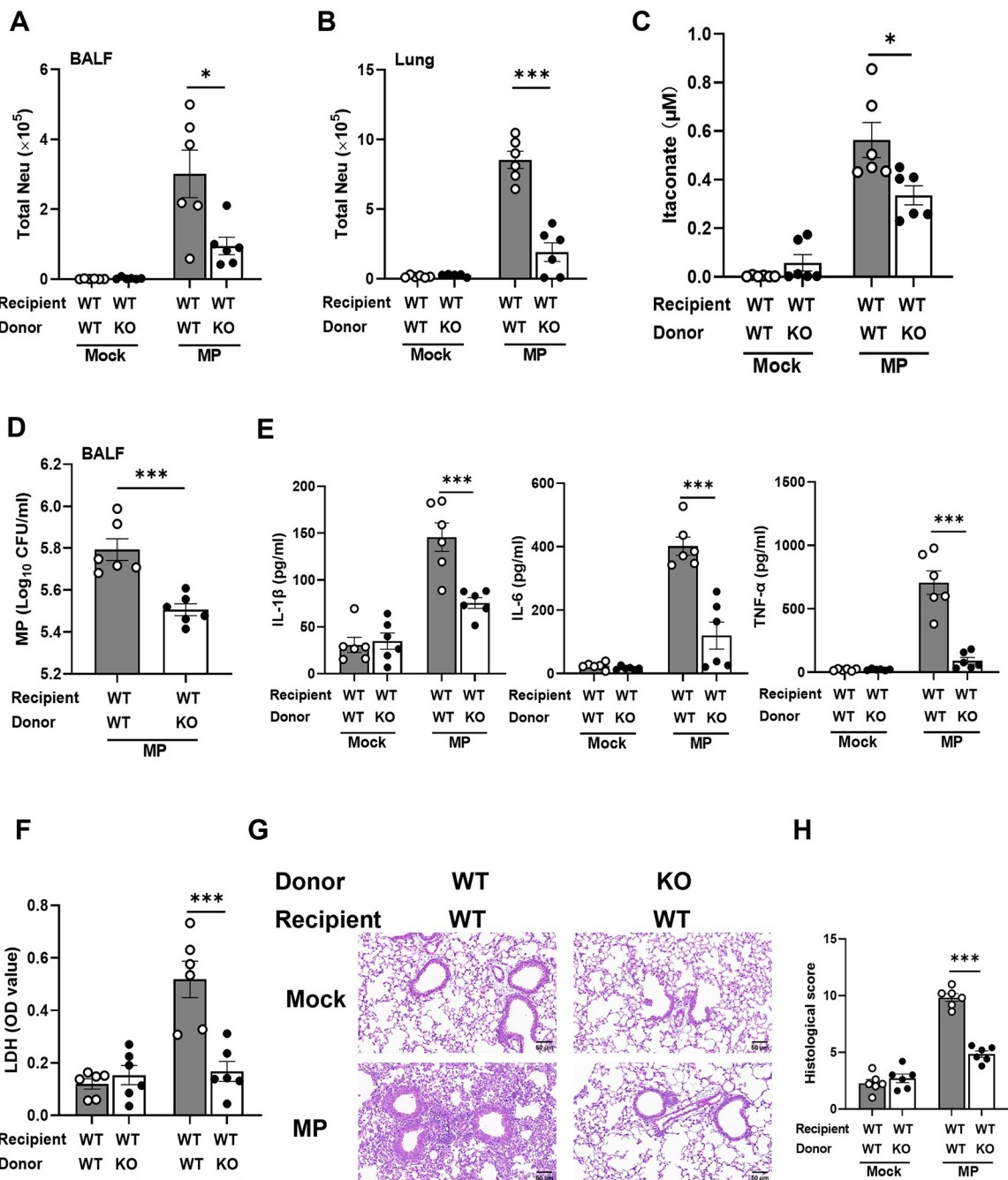

**Fig 3. Adoptive transfer of *Irg1* KO neutrophils attenuates *M. pneumoniae* pneumonia in mice.** Sorted bone marrow (BM) neutrophils from WT and *Irg1* KO mice were transferred into naïve WT mice through the tail vein followed by *M. pneumoniae* (MP) infection. Mice were sacrificed on d3 after *M. pneumoniae* infection (n = 5 or 6 for each mock group and n = 6 for each infection group). Data were pooled from two independent experiments. (A and B) The number Neutrophils in the BALF and in the lung, respectively. (C) Itaconate content in BALF. (D) *M. pneumoniae* burden in BALF. (E) The amount of IL-1β, IL-6 and TNF-α in BALF. (F) LDH levels in the BALF. (G) Representative H&E staining of the lung tissue. Scale bar, 50 μm. (H) Histological scores. Data are presented as mean ± SEM. Statistical significance tested by unpaired, two-tailed Student's *t* test (*, $p < 0.05$; **, $p < 0.01$; ***, $p < 0.001$).

*pneumoniae* infection. We found that β-glucan evidently suppressed mouse neutrophils to express *Irg1* mRNA and IRG1 protein (S6A and S6B Fig), leading to a significant reduction in itaconate production (S6C Fig) *in vitro*. We then intranasally treated mice with β-glucan after

*M. pneumoniae* infection. Indeed, intranasal administration of β-glucan to infected mice significantly reduced *Irg1* mRNA expression in the mouse lung tissue in comparison to infected control mice (Fig 4A). Consistently, IRG1 protein expression in the mouse lung neutrophils determined by flow cytometry exhibited a similar change (Fig 4B). Moreover, the itaconate concentrations in BALF were remarkably reduced in *M. pneumoniae*-infected mice after β-glucan treatment compared to infected control mice (Fig 4C). As expected, β-glucan treatment remarkably reduced neutrophil numbers in BALF and lung tissue (Fig 4D and 4E), *M. pneumoniae* burden (Fig 4F) and LDH levels in BALF (Fig 4G), and the levels of proinflammatory cytokines (IL-1β, IL-6, and TNF-α) (Fig 4H), which led to the remission of lung inflammation (Fig 4I and 4J). Together, these results indicate that *M. pneumoniae* pneumonia can be greatly improved by the IRG1 inhibitor β-glucan.

## Itaconate inhibits neutrophil mtROS production and clearance of *M. pneumoniae*

One of the principal mechanisms of itaconate in modulating the function of myeloid cells is the inhibition of succinate dehydrogenase (SDH)-mediated mitochondrial reactive oxygen species (mtROS) generation [34]. Neutrophil mtROS plays an important role in host defense against bacterial infections [35–37]. To this end, we examined both total ROS (tROS) and mtROS expression in lung neutrophils in WT and *Irg1* KO mice after *M. pneumoniae* infection. Interestingly, we found that *M. pneumoniae* infection resulted in a significant decrease in both tROS and mtROS levels in lung neutrophils from WT mice, but not from *Irg1* KO mice, in comparison to uninfected controls (Fig 5A and 5B). Exogenous administration of itaconate caused a decrease of both tROS and mtROS production in lung neutrophils from *M. pneumoniae*-infected *Irg1* KO mice, comparable to that from infected WT mice (Fig 5A and 5B). Meanwhile, itaconate production did not affect tROS production in alveolar macrophages (S7A Fig). *In vitro*, we infected WT and *Irg1* KO neutrophils with *M. pneumoniae* in the presence or absence of a physiological concentration of itaconate [38] for 12 h, and obtained similar results (S7B and S7C Fig). Moreover, similar to dimethyl malonate, a well-known SDH inhibitor, itaconate significantly suppressed SDH activity in *M. pneumoniae* infected neutrophils (Fig 5C), suggesting that itaconate inhibits neutrophil SDH-mediated mtROS generation. For further confirmation, itaconate inhibited neutrophil oxygen consumption rate (OCR) (Fig 5D), which was not impacted by *M. pneumoniae* infection after 2 h when *Irg1* expression was not induced (S2F and S2G Fig). As itaconate can also inhibit neutrophil nicotinamide adenine dinucleotide phosphate (NADPH) oxidase-derived ROS and neutrophil lipid ROS generation [27,30], we then tested whether itaconate could affect NADPH oxidase activity and lipid ROS in *M. pneumoniae*-infected neutrophils. *M. pneumoniae* infection or itaconate treatment did not affect mouse neutrophil NADPH oxidase activity (S7D Fig), suggesting that itaconate does not impact NADPH-derived ROS in neutrophils during *M. pneumoniae* infection. Moreover, while the ferroptosis inhibitor ferrostatin-1 (Fer-1) suppressed and the ferroptosis activator RSL3 promoted lipid ROS generation in the mouse neutrophils, adding itaconate had no effect on neutrophil lipid ROS generation after *M. pneumoniae* infection (S7E Fig). Therefore, itaconate inhibits neutrophil mtROS production probably through inhibiting SDH activity.

To test whether itaconate impairs neutrophil killing of *M. pneumoniae* via inhibiting mtROS production, we assessed the effect of itaconate, the mtROS inducer mitoparaquat (MitoPQ), the mtROS scavenger MitoTEMPO, and DPI on neutrophil clearance of *M. pneumoniae*. These compounds themselves did not impact *M. pneumoniae* growth *in vitro* (S8A Fig). DPI did not affect the production of mtROS by neutrophils upon *M. pneumoniae*

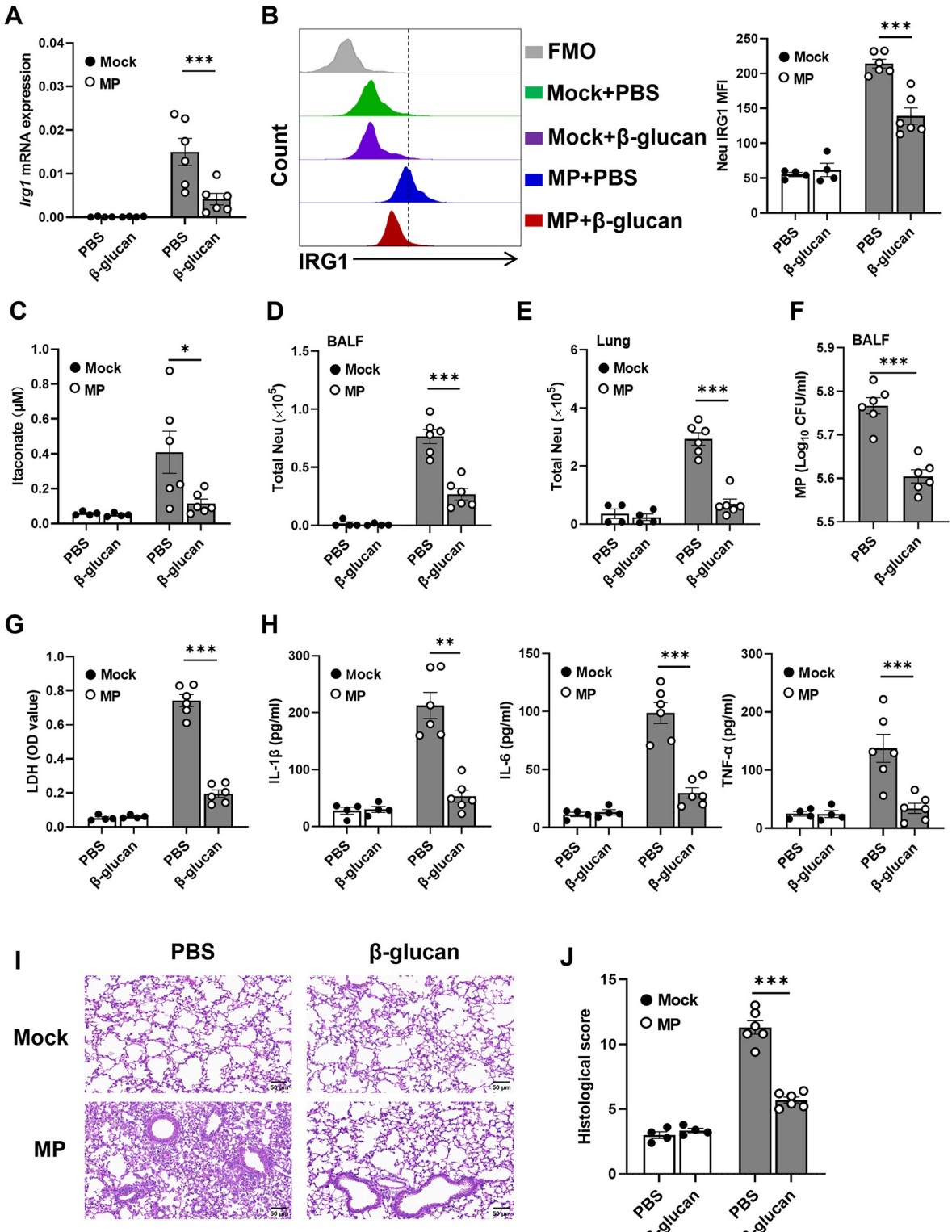

**Fig 4. IRG1 inhibitor β-glucan alleviates *M. pneumoniae* pneumonia in mice.** Mice were intranasally infected with *M. pneumoniae* (MP), followed by intranasally administrated with β-glucan (5 μg/ml in 20 μl PBS/mouse/day) or PBS for 3 consecutive days, mice were sacrificed on day 3 after infection (n = 4 per mock group, n = 6 per infection group). Data were pooled from two independent experiments. (A) *Irg1* mRNA expression in the mouse lung tissue. (B) Flow cytometric analysis of IRG1 expression in lung neutrophils. (C) The amount of itaconate in BALF. (D and E) The number of Neutrophils in BALF and in the lung, respectively. (F) *M. pneumoniae* load in BALF. (G)

LDH levels in BALF. (H) The amount of IL-1β, IL-6 and TNF-α in the BALF. (I) Representative H&E staining of lung tissue. (J) Histological scores. Scale bar, 50 μm. Data are presented as mean ± SEM. Statistical significance tested by unpaired, two-tailed Student's *t* test (*, p<0.05; **, p<0.01; ***, p<0.001).

infection (S8B and S8C Fig). As expected, MitoPQ treatment induced, while MitoTEMPO reduced, mtROS production in the *M. pneumoniae*-infected mouse and human neutrophils (S8B and S8C Fig). Correspondingly, MitoPQ enhanced, while MitoTEMPO impaired, mouse and human neutrophil killing of *M. pneumoniae* (Fig 5E and 5F), indicating that neutrophil mtROS is critical in killing *M. pneumoniae*. Of note, similar to the effect of MitoTEMPO, itaconate treatment decreased neutrophil mtROS production (S8B and S8C Fig), and impaired neutrophil killing of *M. pneumoniae*, whereas DPI treatment did not impact neutrophil bacterial killing (Fig 5E and 5F). Moreover, we obtained similar results by incubation *M. pneumoniae* with corresponding culture supernatant from itaconate, MitoPQ, MitoTEMPO, and DPI-treated neutrophils (Fig 5G), supporting that itaconate impairs neutrophil clearance of *M. pneumoniae* through inhibiting mtROS production. For further confirmation, we sorted WT and *Irg1* KO neutrophils from *M. pneumoniae*-infected mice at day 1 and treated neutrophils with itaconate, MitoPQ, and MitoTEMPO, respectively, followed by *M. pneumoniae* infection. We found that either itaconate or MitoTEMPO treatment impaired *Irg1* KO neutrophil bacterial killing, returning to that of WT neutrophils, whereas MitoPQ significantly enhanced WT neutrophil bacterial killing (Fig 5H). Additionally, similar to previous studies [15,39], *M. pneumoniae* was resistant to neutrophil phagocytosis, and itaconate did not affect neutrophil phagocytosis (S8D Fig). Moreover, itaconate did not affect neutrophil degranulation upon *M. pneumoniae* infection as determined by the membrane CD63 level and neutrophil elastase activity (S8E and S8F Fig). Taken together, itaconate impairs neutrophil clearance of *M. pneumoniae* through inhibiting neutrophil mtROS production.

## Itaconate inhibits neutrophil apoptosis during *M. pneumoniae* infection

As *Irg1* deficiency resulted in a decrease of lung neutrophil numbers after *M. pneumoniae* infection in mice (S1C Fig), we used flow cytometry to determine whether itaconate could affect neutrophil survival. We observed that the percentage of apoptotic lung neutrophils from *M. pneumoniae*-infected *Irg1* KO mice was higher than that from infected WT mice (Fig 6A). Meanwhile, exogenous itaconate treatment cancelled the difference in lung neutrophil apoptosis between WT mice and *Irg1* KO mice after *M. pneumoniae* infection (Fig 6A). We obtained similar results *in vitro* (Fig 6B). We also found that MitoPQ treatment increased, while MitoTEMPO inhibited, neutrophil mtROS production (S9A Fig) and the percentage of apoptotic cells (S9B Fig) compared to vehicle control, suggesting that mtROS is critical in controlling neutrophil apoptosis as described previously [40]. To test whether neutrophil itaconate production induced by *M. pneumoniae* could inhibit neutrophil apoptosis via inhibiting mtROS production, we pre-treated WT neutrophils with MitoPQ, *Irg1* KO neutrophils with MitoTEMPO before *M. pneumoniae* infection and examined neutrophil apoptosis after 12 h. As expected, the addition of MitoPQ increased apoptotic percentage in WT neutrophils, while the addition of MitoTEMPO significantly reduced the apoptotic percentage of *Irg1* KO neutrophils after infection (Fig 6B). Meanwhile, we observed a corresponding change in LDH levels in the supernatant (Fig 6C), the caspase-3 activity (Fig 6D), and mitochondrial potential measured by JC-1 in neutrophils (Fig 6E). Furthermore, WT and *Irg1* KO mice had comparable numbers of lung neutrophils (S9C Fig), as well as comparable mRNA expression of granulopoetic cytokines including *Cxcl1*, and *Cxcl2*, and *Csf3* in the lung tissue at day 1 after *M. pneumoniae* infection (S9D Fig), suggesting that *Irg1* deficiency may not affect neutrophil

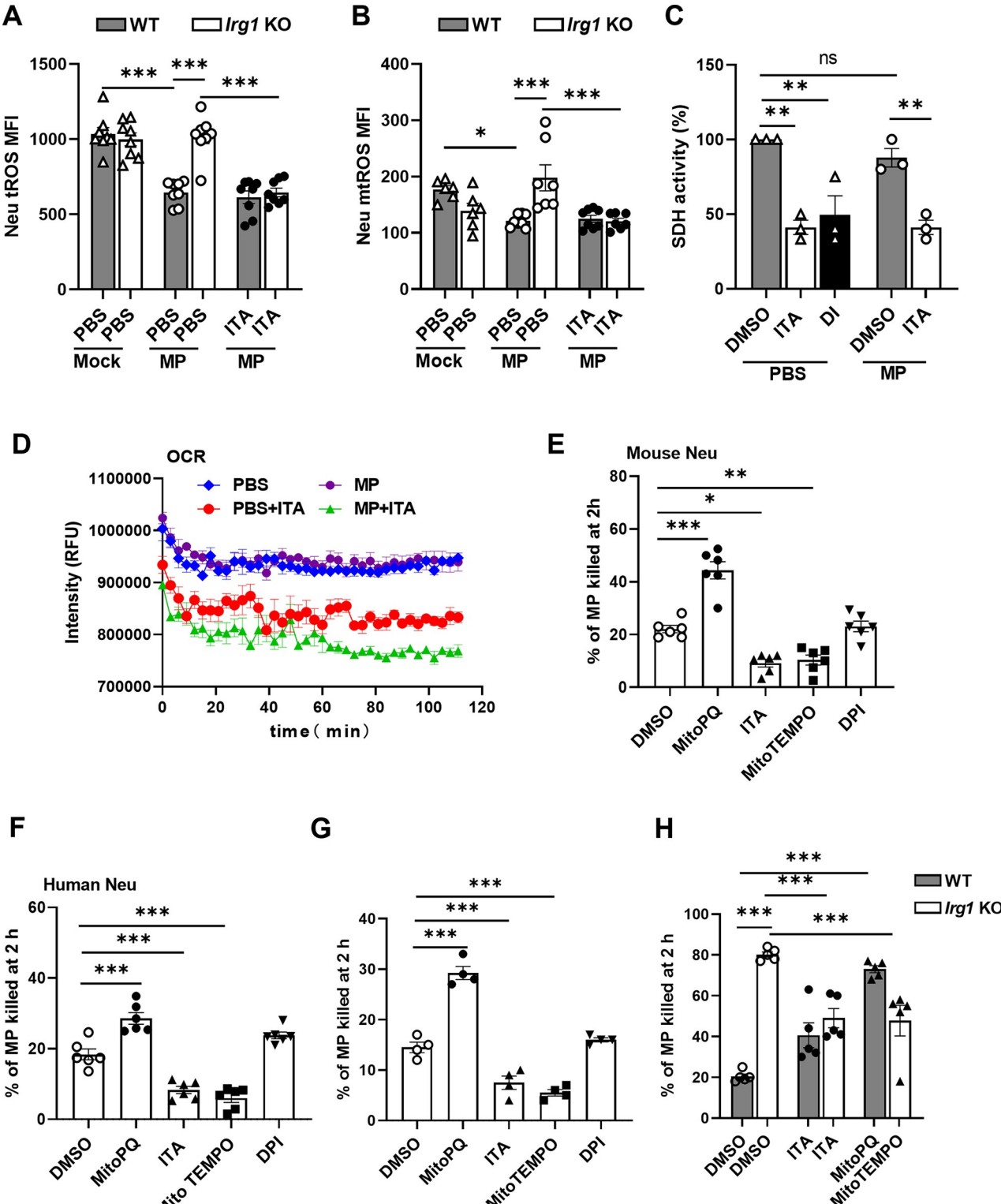

**Fig 5. Itaconate suppresses neutrophil mtROS production and clearance of *M. pneumoniae*.** (A and B) Flow cytometric analysis of lung neutrophil ROS and mtROS gated on CD11b+Ly6G+ cells from WT and *Irg1* KO mice at day 3 after *M. pneumoniae* (MP) infection with or without exogenously itaconate (ITA) treatment (n = 6–8 per group). (C) SDH activity in the mouse BM neutrophils after 2 h of *M. pneumoniae* infection or PBS treatment supplemented with 0 or 5 mM itaconate for 2 h. 10 mM Dimethyl malonate (DI) was used as a positive control. (D) Neutrophil oxygen consumption rate (OCR) during *M. pneumoniae* infection in the presence of 0 or 5 mM itaconate. RFU: relative fluorescence units. (E and F) Bacterial killing by

mouse neutrophils (E) and human neutrophils (F) in the presence of 1 μM MitoPQ, 5 mM itaconate, 100 μM MitoTEMPO, and 10 μM diphenyleneiodonium chloride (DPI). (G) Bacterial killing by incubation *M. pneumoniae* with the culture supernatant from (E). (H) Bacterial killing by sorted neutrophils from *M. pneumoniae*-infected WT and *Irg1* KO mice at day 1 in the presence of 5 mM itaconate, 1 μM MitoPQ, and 100 μM MitoTEMPO as indicated. All the data were pooled from three independent experiments, presented as mean ± SEM. Statistical significance tested by one-way ANOVA test (*, $p < 0.05$; **, $p < 0.01$; ***, $p < 0.001$).

trafficking at the early stage of infection. Previous studies have shown that *M. pneumoniae* infection can induce the production of proinflammatory cytokines in neutrophils [11,14]. We observed that *M. pneumoniae* infection *in vitro* induced comparable mRNA expression of *Il-1β*, *Il-6*, and *Tnf-α* in WT neutrophils and *Irg1* KO neutrophils compared to controls after 12h (S9E Fig), suggesting that itaconate may not impact neutrophils expressing these cytokines during infection. These data indicate that itaconate suppresses neutrophil apoptosis through inhibiting mtROS production during *M. pneumoniae* infection.

Neutrophils mainly depend on glycolysis to exert their function [41]. Itaconate is known to inhibit neutrophil glycolysis during *S. aureus* infection [27]. To investigate whether itaconate could influence neutrophil glycolysis during *M. pneumoniae* infection, we assessed extracellular acidification rate (ECAR) by a fluorescent method. Although *M. pneumoniae* infection increased ECAR in neutrophils, administration of itaconate did not affect the ECAR (S9F Fig), suggesting that itaconate may have no effect on neutrophil glycolysis during infection.

## *M. pneumoniae* induces IRG1 expression in neutrophils through NF-κB and STAT1 signaling pathways involving TLR2

TLR2-mediated signaling pathway is critical for *M. pneumoniae*-induced lung inflammation [12,14,42]. Consistent with previous reports [14], *M. pneumoniae* infection induced TLR2 expression in the neutrophils (Fig 7A). TLR2 signal transduction involves in both NF-κB and STAT1 [14,43–45], which are required for *Irg1* induction during inflammation [46,47]. To investigate the mechanism of induction of *Irg1* in neutrophils by *M. pneumoniae* infection, we measured NF-κB and STAT1 phosphorylation levels in infected neutrophils. Notably, we found that *M. pneumoniae* infection induced phosphorylation of NF-κB and STAT1 in the mouse neutrophils, which was inhibited by TLR2 inhibitor C29, but not by TLR4 inhibitor resatorvid (TAK-242) (Fig 7B). As expected, MyD88 inhibitor TJ-M2010-5 inhibited the activation of NF-κB signaling pathway induced by *M. pneumoniae* (Fig 7B). Moreover, TLR2 agonist Pam3CSK4 TFA also activated NF-κB and STAT1 signaling pathways in mouse neutrophils (Fig 7B), confirming a critical role of TLR2 activation in activating these two signaling pathways. Furthermore, we observed that the expression of *Irg1* mRNA and IRG1 protein induced by *M. pneumoniae* was suppressed by C29, TJ-M2010-5, the NF-κB inhibitor BAY11-7082, and the STAT1 inhibitor fludarabine, but not by resatorvid (TAK-242) (Fig 7C and 7D). Moreover, Pam3CSK4 TFA treatment also significantly induced *Irg1* mRNA and IRG1 protein expression in mouse neutrophils (Fig 7C and 7D). Thus, *M. pneumoniae* induces IRG1 expression in neutrophils through NF-κB and STAT1 signaling pathways involving TLR2.

## Discussion

It is well-known that neutrophil recruitment plays crucial roles in defending against invading pathogen. During *M. pneumoniae* infection, however, lung accumulated neutrophils exert limited killing effects and are the major cells causing inflammatory response or lung injury [7,11,12]. The underlying mechanisms are not fully understood despite the fact that *M. pneumoniae* can escape neutrophil killing by degrading NETs and avoiding phagocytosis [15,16].

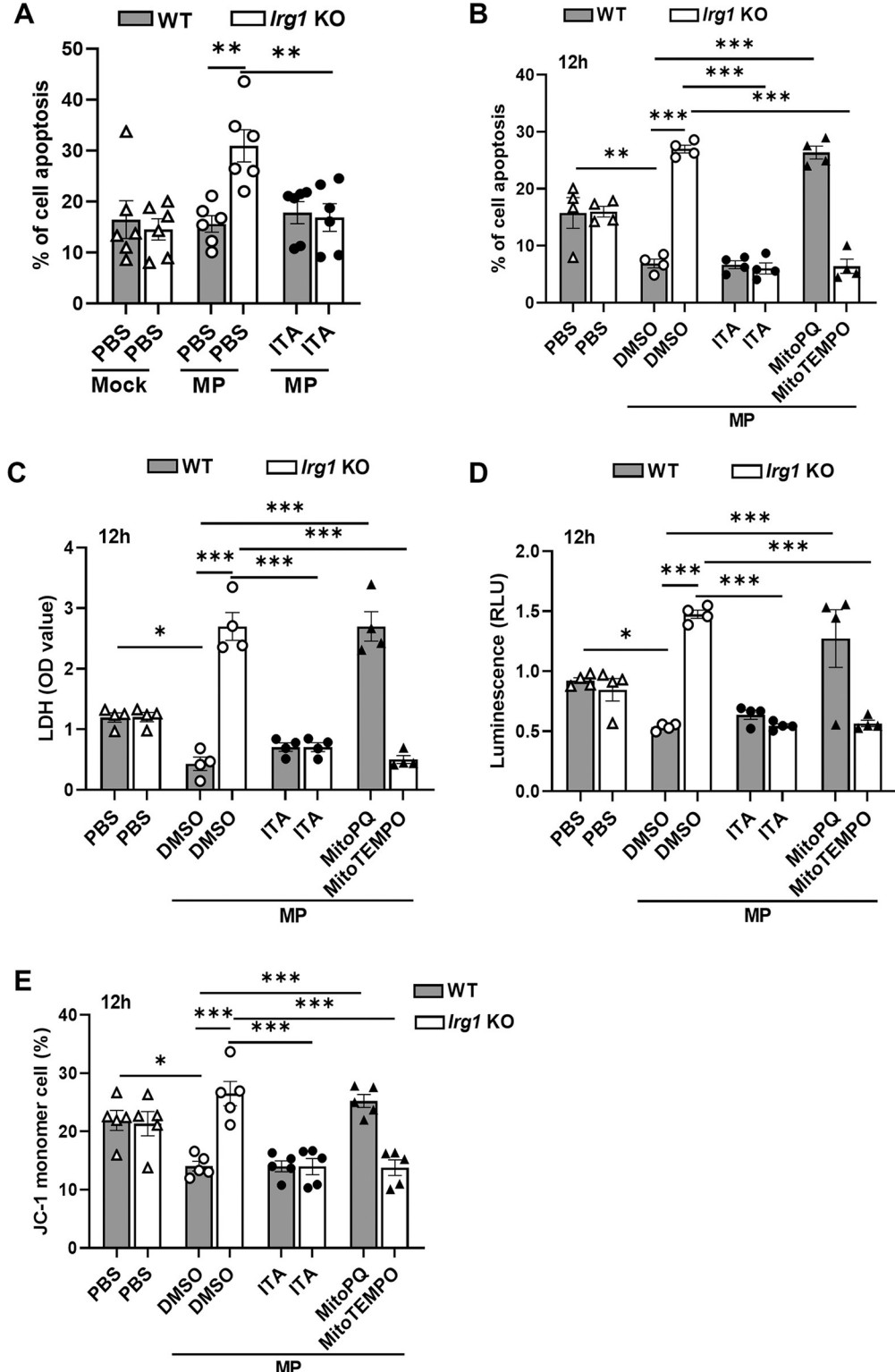

**Fig 6. Itaconate inhibits neutrophil apoptosis during *M. pneumoniae* infection.** (A) Flow cytometric analysis of the apoptosis of CD11b+Ly6G+ lung neutrophil from WT and *Irg1* KO mice at day 3 after *M. pneumoniae* (MP) infection with or without exogenously itaconate (ITA) treatment (n = 6 per group). (B-E) BM neutrophils from WT and *Irg1* KO mice were pre-treated with 5 mM itaconate, 1 μM MitoPQ, or 100 μM MitoTEMPO as indicated for 1 h follow by *M. pneumoniae* infection (MOI = 10) for 12 h. (B) The proportion of apoptotic cells by flow cytometry. (C) LDH levels

in the culture supernatant. (D) Caspase-3 activity in neutrophils. (E) JC-1 monomers in neutrophils by flow cytometry. All the data were pooled from three independent experiments, presented as mean ± SEM. Statistical significance tested by one-way ANOVA test (*, $p < 0.05$; **, $p < 0.01$; ***, $p < 0.001$).

In this study, we reveal a critical role for the *Irg1*/itaconate metabolic pathway in the pathogenesis of *M. pneumoniae* pneumonia. We show that *M. pneumoniae* infection mainly induces neutrophil itaconate production, which dampens neutrophil bactericidal activity and impedes neutrophil apoptosis through inhibiting mtROS. Deficiency of *Irg1*, adoptive transfer of *Irg1* KO neutrophils, or administration of β-glucan (which inhibits *Irg1* expression) can significantly attenuate *M. pneumoniae* pneumonia in mice. Mechanically, we show that *M. pneumoniae* induces *Irg1* expression by activating NF-κB and STAT1 signaling pathways involving TLR2. Our findings thus uncover a novel mechanism by which *M. pneumoniae* evades host innate immunity.

Although the role of itaconate-producing macrophages in inflammation and infection has been extensively studied [20,48], emerging evidence suggests that neutrophil itaconate production also plays a critical role in regulating inflammatory response and host immunity [27, 29, 30]. By using a mouse model of *M. pneumoniae* infection, we found that neutrophils were the main cells expressing *Irg1* and producing itaconate as depletion of neutrophils *in vitro* and *in vivo* dramatically decreased IRG1 expression. Importantly, we demonstrated that itaconate limited neutrophil clearance of *M. pneumoniae* by inhibiting neutrophil mtROS generation, suggesting a critical role of neutrophil mtROS in host antimicrobial function as described previously [35,36]. Inhibition of SDH or SDH-mediated mtROS production is one of key mechanisms of itaconate in regulating macrophage function and in mediating immune tolerance [28,34]. We also found that itaconate inhibited SDH activity in neutrophils, which probably explains the inhibition of mtROS production by itaconate during *M. pneumoniae* infection. Furthermore, itaconate treatment decreased neutrophil OCR, which was not impacted by *M. pneumoniae* infection within 2 h when *Irg1* expression was not induced, supporting the inhibition of mitochondrial oxidative phosphorylation and mtROS production by itaconate. Recently, in a mouse model of *S. aureus* lung infection, Tomlinson et al. showed that itaconate produced by neutrophils can inhibit neutrophil oxidative burst [27]. By contrast, we showed that neither *M. pneumoniae* infection nor itaconate treatment affected the activity of NADPH oxidase in neutrophils. The reason why *M. pneumoniae* did not cause neutrophil oxidative burst may be due to its resistance to phagocytosis (S8D Fig) and as previously described [15]. Moreover, *Irg1* deficiency in myeloid cells including monocytes/macrophages, neutrophils, and some dendritic cells, exacerbates *Mtb*-induced lung pathology and disease, suggesting a protective role of itaconate during *Mtb* infection [25]. These results and ours indicate that itaconate-producing neutrophils have different effects in different microbial pulmonary infections. Macrophages play an important role in eliminating *M. pneumoniae* [8]. We also found that *M. pneumoniae* induced IRG1 expression in alveolar macrophages; however, *Irg1* deficiency did not affect alveolar macrophage ROS production. Whether itaconate could regulate monocyte/macrophage function during *M. pneumoniae* infection remains to be investigated.

In addition, we showed that itaconate acted as a pro-inflammatory metabolite during *M. pneumoniae* infection as *Irg1* deficiency remarkably mitigated *M. pneumoniae* pneumonia in mice at day 3 p.i. characterized by the decreased numbers of lung neutrophils and the reduced levels of pro-inflammatory cytokines in BALF. Meanwhile, the beneficial effects were abolished by intranasally administration of itaconate to *Irg1* KO mice. Notably, we found that supplementation with exogenous itaconate did not lead to more severe lung inflammation in WT mice, suggesting that the effects of itaconate may not be dose-dependent or there may be a

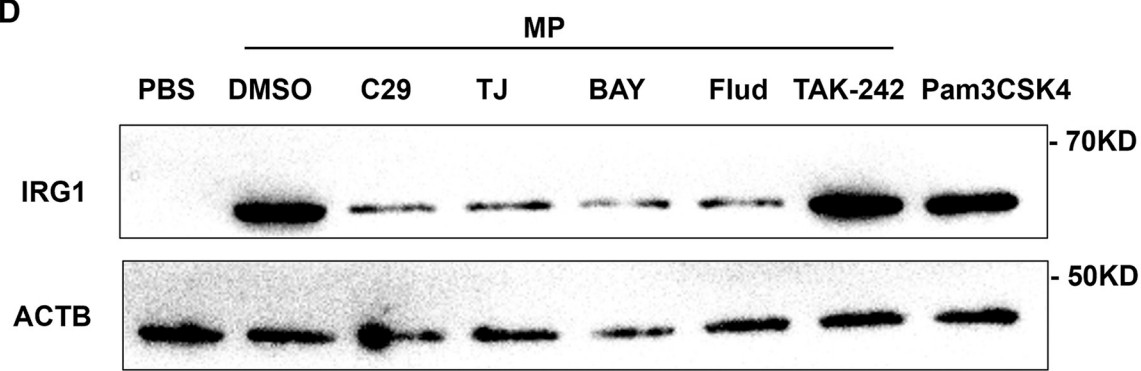

**Fig 7. *M. pneumoniae* induces *Irg1* expression in neutrophils through activating NF-κB and STAT1 signaling pathways.** (A) TLR2 expression on mouse BM neutrophils after 12 h of infection by *M. pneumoniae* (MP) (MOI = 0 or 10) by flow cytometry. The experiment was performed three times independently. Data are presented as mean ± SEM. Statistical significance tested by unpaired, two-tailed Student's *t* test. (B) Western blot analysis of the phosphorylation and total protein expression of p65 and STAT1 in mouse BM neutrophils after 1 h of infection with *M. pneumoniae* (MOI = 0 or 10) or TLR2 agonis Pam3CSK4 TFA (1 µg/ml) treatment; Cells were pretreated with TLR2 inhibitor C29

(100 μM), MyD88 inhibitor TJ-M2010-5 (TJ, 10 μM), TLR4 inhibitor TAK-242 (100 μM) for 1 h before infection. The experiment was performed three times independently. (C and D) Mouse BM neutrophils were pretreated with C29 (100 μM) and TJ-M2010-5 (TJ, 10 μM), BAY11-7082 (BAY, 2 μM), Fludarabine (Flud, 10 μM), TLR4 inhibitor TAK-242 (100 μM), for 1 h followed by *M. pneumoniae* infection (MOI = 0 or 10) or Pam3CSK4 TFA (Pam3CSK4, 1 μg/ml) for 12 h as indicated. (C) *Irg1* mRNA expression in neutrophils, pooled from three independent experiments. Statistical significance tested by one-way ANOVA test. (D) IRG1 protein expression in neutrophils, representative of three experiments. Data are presented as mean ± SEM. (*, $p < 0.05$; **, $p < 0.01$; ***, $p < 0.001$).

certain threshold. This indicates that the endogenous itaconate production induced by *M. pneumoniae* infection may be sufficient to evoke lung damage. Moreover, the mRNA expression levels of *Il-1β*, *Il-6*, and *Tnf-α* were found to be comparable in WT and *Irg1* KO neutrophils after *M. pneumoniae* infection *in vitro*. These data support that IRG1-expressing neutrophils are inflammatory during *M. pneumoniae* infection, similar to that following trauma [29]. Notably, we found that itaconate inhibited neutrophil apoptosis during *M. pneumoniae* infection via inhibiting neutrophil mtROS production, which may lead to exacerbation of neutrophil-mediated lung inflammation and injury. Meanwhile, our results support that mtROS exerts a critical role in inducing neutrophil apoptosis as described previously [40]. In contrast to our results, in a mouse model of *S. aureus* infection, Tomlinson et al. showed that itaconate impaired neutrophil survival by inhibiting glycolysis [27]. However, our data showed that itaconate did not affect neutrophil glycolysis, likely due to the resistance of phagocytosis of *M. pneumoniae*. Furthermore, although we did not see a notable change of lung neutrophil numbers, as well as the mRNA expression of *Cxcl1*, and *Cxcl2*, and *Csf3* in the lung tissue, between WT and *Irg1* KO mice at day 1 p.i., whether itaconate could impact neutrophil migration during *M. pneumoniae* infection warrants further investigation. Also, in most experiments, mice were sacrificed at day 3 p.i. when itaconate levels and neutrophil numbers were high and lung inflammation was pronounced. Further experiments are required to evaluate whether myeloid-derived itaconate would affect host immunity against *M. pneumoniae* infection at later stages.

It is well-known that activation of TLR2/MyD88/NF-κB signaling pathway in myeloid cells is essential in eliciting inflammatory response upon *M. pneumoniae* infection [14,49]. Interestingly, we found that this pathway was also required for *Irg1* expression in neutrophils, supporting that TLR2 activation induces *Irg1* expression in myeloid cells [46,50]. Meanwhile, we also found that both heat-killed *M. pneumoniae* and *M. pneumoniae*-derived LAMP can also induce *Irg1* expression in neutrophils, suggesting that *M. pneumoniae* may induce *Irg1* expression via its LAMP. As IRG1-induced itaconate is detrimental to host during *M. pneumoniae* infection, our data thus, from metabolic aspect, support that mycoplasma LAMP is a potent inducer of inflammatory response as described previously [51–53]. Additionally, we found that *M. pneumoniae* induced the activation of STAT1 signaling involving TLR2, which was also required for *Irg1* expression. Although STAT1 signaling involves in *Irg1* expression in myeloid cells [47], the precise mechanism of the activation of STAT1 by *M. pneumoniae* as well as the interaction between STAT1 signaling and NF-κB signaling requires further investigation.

Neutrophil accumulation in the lungs is a hallmark feature of *M. pneumoniae* pneumonia in children [6,7,10,13]. Importantly, we observed a positive correlation between neutrophil counts and itaconate concentrations in the BALF from patients with severe *M. pneumoniae* pneumonia. Meanwhile, neutrophils had a higher level of IRG1 expression than non-neutrophils in BALF from severe patients with *M. pneumoniae* infection, suggesting that neutrophils may also be the main source of itaconate during *M. pneumoniae* infection in humans. Notably, the concentration of itaconate in the BALF of human patients with *M. pneumoniae* pneumonia was much lower than that in infected mice (20 nM vs. 1000 nM). Meanwhile, mouse

neutrophils produced more itaconate than human neutrophils after *M. pneumoniae* infection *in vitro* (S2H and S2J Fig). This apparent lower capacity of human IRG1 to produce itaconate is in line with previous reports for human macrophages [54,55], and may be due to the reduced enzymatic activity as described previously [56]. Despite this discrepancy, itaconate impaired human neutrophil bacterial killing by inhibiting mtROS production upon *M. pneumoniae* infection, similar to the results in mouse neutrophils. These results suggest that inhibition of human neutrophil itaconate production may exert a protective effect in *M. pneumoniae*-infected patients.

In summary, we demonstrate that activation of the NF-κB and STAT1 signaling pathways by *M. pneumoniae* induces neutrophil expressing *Irg1* and producing itaconate, which impairs neutrophil clearance of *M. pneumoniae* and suppresses neutrophil apoptosis by inhibiting mtROS generation. Suppressing neutrophil itaconate production enhances neutrophil killing of *M. pneumoniae* and reduces pathological inflammatory responses. Thus, our findings suggest that pharmacological agents that inhibit *Irg1*/itaconate metabolic pathway in neutrophils might be effective in treating *M. pneumoniae* pneumonia in humans.

## Materials and methods

### Ethics statement

Our study protocol was approved by the Ethics Committee at the First Affiliated Hospital, Hengyang Medical School, University of South China (approval no. 2023-K-EK-44-01). Written informed consents were obtained from all participants. All mouse experimental procedures were approved by the Institutional Animal Use and Ethics Committee of University of South China (approval no. USC2024XS019).

### Mice

*Irg1*(also known as *Acod1*) KO C57BL/6J mice (strain no. T012677) were purchased from GemPharmatech (Nanjing, China). C57BL/6J-Ly5.1 (CD45.1) mice were provided by Prof. Jie Zhou from Tianjin Medical University. C57BL/6J and BALB/c mice were purchased from Hunan SJA Laboratory Animal Co, Ltd. (Animal Production License no. SCXK 2019–0004). All mice were used at 6–8 weeks old, and an equal ratio of male: female animals were randomly assigned to cages.

### Reagents

The following compounds were used in this study: Itaconate (Sigma-Aldrich, I29204), β-D-glucan (Sigma-Aldrich, G6513), mtROS inducer mitoparaquat (MitoPQ, MCE, HY-130278), mtROS scavenger Mito-TEMPO (MCE, HY-112879), dipheny leneiodonium chloride (DPI, MCE, HY-100965), ferrostatin-1 (Fer-1, MCE, HY-100579), ferroptosis activator (1S,3R)-RSL3 (RSL3, MCE, HY-100218A), dimethyl malonate (DI, Sigma-Aldrich, 136441), STAT1 inhibitor Fludarabine (MCE, HY-B0069), BAY 11–7082 (MCE, HY13453), TLR2 inhibitor C29 (Selleck, S6597), MyD88 inhibitor TJ-M2010-5 (MCE, HY-139397), TLR4 inhibitor TAK-242 (Selleck, S7455), and TLR2 agonist Pam3CSK4 TFA (MCE, HY-P1180A). The pH of the medium supplemented with itaconate was corrected to 7.0 with 10N NaOH as described [27].

### Clinical sample collection

Bronchoalveolar lavage fluid (BALF) samples (n = 21) were collected from patients with severe *M. pneumoniae* pneumonia. Demographic and clinical information of all the patients were shown in S1 Table. The criteria for patients with severe *M. pneumoniae* pneumonia were: 1)

Children with obvious symptoms of pneumonia including cough, fever $\geq 38.5°$C, breathless with respiratory rate $< 50$ breaths/min at age $\geq 3$ years old, chest retractions, abnormal chest computed tomography (CT) findings and 2) *M. pneumoniae* DNA was detected in BALF and throat swabs by real-time PCR, and serum *M. pneumoniae* IgM were detected by enzyme-linked immunosorbent assays (ELISA). Chest imaging score was evaluated according to the following criteria: non-consolidation, consolidation, consolidation and a small amount of pleural effusion, and consolidation and a medium to large amount of pleural effusion as described previously[57]. Cases were excluded if they were infected with other pathogens and diagnosed as bronchopulmonary malformation, chronic disease, cardiovascular disease, immune deficiency, and immune suppression.

## *M. pneumoniae* and infection mouse model

As described previously [58], *M. pneumoniae* (strain M129; ATCC 29342) was reconstituted in Pleuropneumonia-like organism (PPLO) broth (BD Biosciences, USA) for 48–72 h and transferred to a flask containing 20 ml of PPLO broth at $37°$C until the color of the broth changed to orange hue (about 5 days). Then, the supernatant was decanted, and 4 ml of fresh PPLO broth was added to the flask followed by harvesting the adherent *M. pneumoniae* from the bottom of the flask using a cell scraper. *M. pneumoniae* suspension was centrifuged at $4°$C at 6000 g for 30 min and resuspended in PPLO broth. The colony-forming unit (CFU) of *M. pneumoniae* was measured by solid culture and counted the colonies under a microscope as described [59]. For *in vivo* infection, BALB/c mice were infected intranasally with $10^7$ colony-forming units (CFU) *M. pneumoniae* in 50 μl PPLO medium as described [9,32]. C57BL/6J wild-type (WT) and *Irg1* KO mice were inoculated intranasally with 50 μl of *M. pneumoniae* containing $10^8$ CFU as described [60]. Mock inoculations were with 50 μl PPLO medium alone. For *in vitro* assays, neutrophils were infected with *M. pneumoniae* with a multiplicity of infection (MOI) = 10 in PBS.

## Extraction of *M. pneumoniae* LAMP

*M. pneumoniae* LAMP was prepared as described previously [61]. Briefly, *M. pneumoniae* pellet was resuspended in 5 mL Tris-EDTA buffered solution (50 mM Tris pH 8.0, 0.15 M NaCl, 1 mM EDTA), followed by adding Triton X-114 to a final concentration of 2%, and incubated at $4°$C for 1 h. Then, the Triton X-114 lysate was incubated at $37°$C for 10 min for phase separation. The upper aqueous phase was then removed and replaced with the same volume of Tris-EDTA buffered solution. The phase separation process was repeated twice. The final Triton X-114 phase was resuspended to its original volume in Tris-EDTA buffered solution at $4°$C. Then, 2.5 volumes of ethanol were added to precipitate the proteins at -$20°$C for 12 h. The LAMP was obtained by centrifuging at 14000 rpm for 20 min and was resuspended in PBS by sonication for 3 min. Protein content was determined by bicinchoninic acid (BCA) assay.

## Neutrophil depletion and adoptive transfer

For neutrophil depletion *in vivo*, C57 WT mice were intraperitoneally injected with 0.2 mg ultra-LEAF purified anti-mouse Ly-6G (Biolegend, 127649) or ultra-LEAF purified rat IgG2a, κ isotype control (Biolegend, 400565) at day 1 after *M. pneumoniae* infection. Mice were sacrificed at day 3 p.i. For adoptive transfer of neutrophils, 3 million sorted bone marrow (BM) neutrophils from WT mice or *Irg1* KO mice were intravenously transferred into recipient mice 30 min prior to *M. pneumoniae* infection. Mice were sacrificed at day 3 p.i.

## Cell isolation from tissue

Mouse BM cells, BALF, and lung leukocytes were obtained as described previously [62,63]. Briefly, BM cells were collected by flushing femurs and tibias with a syringe containing RPMI 1640 media. Red blood cells were lysed by using ammonium-chloride-potassium (ACK) buffer. BALF was obtained by flushing the lungs with 0.8 ml of cold PBS twice via a thin tube inserted into a cut made in the trachea. The right upper lung lobe was preserved in 4% paraformaldehyde and paraffin-embedded tissue sections were stained with hematoxylin and eosin (H&E). H&E images per mouse were blindly scored based on the severity of peribronchiolar/peribronchiolar infiltrates (including quantity and quality), bronchiolar/luminal exudates, perivascular infiltrates, and parenchymal pneumonia as described previously [33,64]. Lung cells were obtained by mechanical disruption on 70 μm cell strainers and the remaining red blood cells were lysed with ACK buffer. Lung leukocytes were collected by centrifugation in 35% Percoll (Cytiva) at 700 g for 15 min.

## Purification of neutrophils

Lung cells or bone marrow cells were first labeled with biotinylated anti-mouse Ly-6G (Biolegend, 127604) followed by streptavidin-paramagnetic particles (BD Bioscience, 127604) as described previously [63]. Neutrophils were positively selected using a BD IMag Cell Separation Magnet. Human neutrophils were isolated from human peripheral blood by using a human blood neutrophil isolation kit (TBD, LZS11131, China) according to manufacturer's instructions. The purity and viability of obtained neutrophils were > 90%.

## Flow cytometry

Single-cell suspensions were incubated with an unlabeled purified anti-Fc receptor blocking antibody (anti-CD16/CD32) (BD Biosciences, 553141) before staining with the appropriated combination of fluorochrome-conjugated antibodies. Cell surface molecule staining was performed at 4°C for 30 min in PBS in the dark. For intracellular IRG1 staining, lung leukocytes were fixed and permeabilized using an intracellular fixation and permeability kit (eBioscience, 88-8824-00) followed by antibody staining according to manufacturer's instructions. Cells were acquired on a FACS Calibur flow cytometer (BD Biosciences). Flow data were analyzed using FlowJo software (Tree Star Inc.). The antibodies used in this study are fixable viability Dye eFluor 660 (Invitrogen, 65-0864-14), FITC anti-mouse/human CD11b (Biolegend, 101206), PerCP-Cy5.5 anti-mouse/human CD11b (Biolegend, 101227), APC anti-mouse/human CD11b(Biolegend, 101212), PE anti-mouse Ly-6G(Biolegend, 127608), APC anti-mouse Ly-6G (Biolegend, 127614), PerCP-Cy5.5 anti-mouse Ly6C (eBioscience, 45-5932-80), PerCP-Cy5.5 anti-mouse CD45 (Biolegend, 103132), FITC anti-mouse CD45 (Biolegend, 103108), PE anti-mouse CD45 (Biolegend, 103106), APC anti-mouse CD11c (Biolegend, 117310), PE anti-mouse Siglec-F (BD Biosciences, 552126), APC anti-mouse CD63 (Biolegend, 143905), FITC anti-mouse TLR2 (Invitrogen, 119021–82), FITC-Streptavidin (Biolegend, 405202), IgG Isotype (Biolegend, 400405), PE anti-mouse CD45.1 (Invitrogen, 12-0453-91), APC anti-mouse CD45.2 (BD Biosciences, 561875), APC anti-human CD15 (Invitrogen, 17-0158-42), PE anti-human CD16 (Invitrogen, 12-0168-42), and PerCP-Cy5.5 anti-human CD45 (Invitrogen, 45-0459-42).

## Quantitative real-time PCR

Total RNA was extracted from cells using RNAiso Plus (Takara) and reverse transcribed with a synthesis kit (Takara, RR036A). Gene expression was then measured by qRT-PCR with TB

Green (Takara, RR820A). Normalized gene expression values were calculated as the ratio of expression of mRNA of interest to the expression of mRNA for *β-actin* by the $2^{-\Delta Ct}$ method. Primer sequences used in this study are: mouse *β-actin* forward: 5′-CGT GCG TGA CAT CAA AGA GAA G-3′; mouse *β-actin* reverse: 5′-CGT TGC CAA TAG TGA TGA CCT G-3′; mouse *Irg1* forward: 5′-GTT TGG GGT CGA CCA GAC TT-3′; mouse *Irg1* reverse: 5′-CAG GTC GAG GCC AGA AAA CT-3′; mouse *Cxcl1* forward: 5′-ACT CAA GAA TGG TCG CGA GG-3′; mouse *Cxcl1* reverse: 5′-GTG CCA TCA GAG CAG TCT GT-3′; mouse *Cxcl2* forward: 5′-GGC AAG GCT AAC TGA CCT GG-3′; mouse *Cxcl2* reverse: 5′-CTC AGA CAG CGA GGC ACA TC-3′; mouse *csf3* forward: 5′-TAT AAA GGC CCC CTG GAG CTG-3′; mouse *csf3* reverse: 5′-GCT GCA GGG CCA TTA GCT TC-3′; mouse *Il-6* forward: 5′-TAC CAC TTC ACA AGT CGG AGG C-3′; mouse *Il-6* reverse: 5′-CTG CAA GTG CAT CAT CGT TGT TC-3′; mouse *Tnf-α* forward: 5′-GGT GCC TAT GTC TCA GCC TCT T-3′; mouse *Tnf-α* reverse: 5′-GCC ATA GAA CTG ATG AGA GGG AG-3′; mouse *Il-1β* forward: 5′-TGG ACC TTC CAG GAT GAG GAC A-3′; mouse *Il-1β* reverse: 5′-GTT CAT CTC GGA GCC TGT AGT G-3′; Human *Gapdh* forward: 5′-ACA GCC TCA AGA TCA TCA GC-3′; Human *Gapdh* reverse: 5′-GGT CAT GAG TCC TTC CAC GAT-3′; Human *Irg1* forward: 5′-TGG TTC ACT CCT CCT GAA CTG-3′; Human *Irg1* reverse: 5′-TCT ATC CCA TGC TTG GCA GC-3′.

## Western blotting

Total cell lysates were prepared using a Radio-Immunoprecipitation Assay (RIPA) Lysis Buffer (Strong) (CWBIO, CW2333S), and a cocktail of protease inhibitors (CWBIO, CW2200S). The proteins were separated by SDS-PAGE, followed by transferred to a polyvinylidene difluoride (PVDF) membrane (Millipore, ISEQ00010). The targeted molecules were probed using specific primary antibodies overnight at 4°C and HRP-conjugated secondary antibodies at room temperature, the protein bands were detected with a SuperKine West Femto Maximum Sensitivity Substrate (Abkine, BMU102). The images were captured using a ChemiScope imaging system (SYNGENE, Australasia). The primary antibodies used are anti-IRG1 (Abcam, ab222411), NF-κB p65(CST, 8242), anti-pSTAT1 (Tyr 701) (Santacruz, SC-7988), Phospho-NF-κB p65 (Ser536) (CST, 3033), anti-STAT1 (CST, 9172), and anti-β-actin (ZENBIO, T200068-8F10).

## Ultra-performance liquid chromatography-tandem mass spectrometry

Itaconate in BALF or culture supernatant was quantified by ultra-performance liquid chromatography-tandem mass spectrometry (UPLC-MS/MS) (Metabo-Profile, Shanghai, China). Briefly, 20 μl sample was transferred to a 96-well plate and put it to the Eppendorf epMotion workstation (Eppendorf Inc., Humburg, Germany). Then 120 μl of cold methanol (with internal standard) was added to the wells and vigorously vortexed for 5 min, followed by centrifuging at 4000 g at 4°C for 30 min. After that, 20 μl of derivatization reagent was added to the wells and inoculated at 30°C for 60 min. The mixture was then diluted with 330 μl cold 50% methanol and was centrifuged at 4000 g at 4°C for 30 min. Subsequently, 135 μl of supernatant was transferred to a new 96-well plate with 10 μl internal standards and finally subjected to LC-MS analysis. To ensure data reproducibility, a reagent blank and the quality control samples were added with the test samples and injected regularly throughout the process.

## Enzyme-linked immunosorbent assay

Cytokines in BALF were measured by ELISA by using the kits of IL-1β (Biolegend, 432604), IL-6 (Biolegend, 431304), and TNF-α (Biolegend, 430904) following the manufacturer's instructions.

## ROS measurement

Total ROS production was measured by using CM-H2DCFDA (Invitrogen, C6827) according a protocol provided by the manufacturer. To determine mtROS production, cells were incubated with HBSS containing 1 μM MitoSOX Red (Yeasen, China, 40778ES50) at 37˚C for 30 min. To measure lipid ROS, cells were treated with PBS containing 2 μM C11 BODIPY 581/591 (Cayman, USA, 27086) and were incubated at 37˚C for 30 min. After incubation with these dyes, cells were washed twice and ROS production was detected by flow cytometry.

## Measurement of neutrophil apoptosis

The percentage of neutrophil apoptosis was determined by flow cytometry using Annexin V-FITC (APExBIO, USA, K2003) and 7-AAD (Invitrogen, 00-6993-50) following the manufacturer's instructions. For some *in vitro* assays, neutrophil apoptosis was verified by measuring the caspase 3 activity with a Caspase-Glo 3/7 Assay kit (Promega (Beijing) Biotech Co., Ltd, China, G8091).

## Mitochondrial membrane potential detection

Mitochondrial membrane potential was determined by using a fluorescent probe JC-1 (5,5',6,6'-tetrachloro-1,1',3,3'-tetraethylbenzimidazolylcarbocyanine iodide) (Beyotime, China, C2006). Neutrophils were stained with 1 μM JC-1 dye at 37˚C for 30 min, washed and immediately detected by flow cytometry.

## Lactate dehydrogenase assay

The lactate dehydrogenase (LDH) levels in BALF and the cell supernatant were determined using a Cytotoxicity LDH assay kit (Invitrogen, C20300) according to the manufacturer's instructions.

## Neutrophil NADPH oxidase activity assay

Neutrophils were seeded in 96-well plates at a density of $5 \times 10^5$ cells per well in RPMI 1640 complete medium supplemented with 0, 2.5 or 5 mM itaconate. Control wells were supplemented with 10 μM of the NADPH oxidase inhibitor diphenyleneiodonium chloride (DPI) or 1 mM of the NADPH oxidase activator phorbol myristate acetate (PMA). After 1 h incubation at 37˚C, the cells in some wells were infected with *M. pneumoniae* at a MOI of 10. After 2 h infection at 37˚C, the activity of NADPH oxidase in cells was measured by using NADPH oxidase kit (Suzhou Grace Biotechnolgy Co., Ltd, China, G0138W).

## SDH activity assay

Neutrophils were seeded in 96-well plates at a density of $5 \times 10^5$ cells per well in RPMI 1640 complete medium supplemented with 5 mM itaconate for 1 h at 37˚C, followed by the infection with *M. pneumoniae* at a MOI of 10. 10 mM dimethyl malonate (DI, Sigma-Aldrich) was used as positive controls. After incubation for 2 h at 37˚C, SDH activity in neutrophils was determined by using a MitoCheck complex II activity assay kit (Cayman, USA, 700940) according to the manufacturer's instructions. For calculations, we first plotted data as absorbance (y-axis) vs time (x-axis), and calculated the slope for the linear portion of the curve to determine the reaction rate, followed by calculating SDH activity (%) using (Rate of Sample wells/Rate of Control wells) ×100.

## Neutrophil bacterial killing assay

Neutrophil bacterial killing assay was performed as described previously [27]. Briefly, neutrophils were seeded in 96-well plates at a density of $5\times10^5$ cells per well in RPMI 1640 complete medium supplemented with 5 mM itaconate, and then infected with *M. pneumoniae* at a MOI of 10. The same number of *M. pneumoniae* was added to control wells in the absence of neutrophils. After 2 h of infection at 37˚C, a 100 µl sample from each well was mixed with 900 µl of water +0.1% NP-40 (Thermo Fisher, YD366404). The samples were plated on a PPLO agar plate. After 1 week incubation at 37˚C, colonies were counted and bacteria killing was evaluated relative to the wells without neutrophils.

## Neutrophil degranulation

Neutrophils were seeded in 96-well plates at a density of $5\times10^5$ cells per well in RPMI 1640 complete medium supplemented with 5 mM itaconate for 1 h at 37˚C, and then infected with *M. pneumoniae* at a MOI of 10, or treated with 10 µg/ml of cytochalasin D (Yeasen, 53215ES03) for 5 min before infection. The infection was synchronized by centrifuging the plates at 500 g for 2 min. After 2 h infection at 37˚C, cells were incubated with anti-CD63 surface antibodies (Biolegend, 143905) and detected by flow cytometry as described previously [65].

## Neutrophil elastase activity assay

Neutrophils were seeded in 96-well plates at a density of $5\times10^5$ cells per well in RPMI 1640 complete medium supplemented with 5 mM itaconate for 1 h at 37˚C, and then infected with *M. pneumoniae* at a MOI of 10. After 2 h of infection, the cells were treated with $1 \times$ protease inhibitor (MCE, HY-K0010) for 30 min, and were lysed with 0.1% Triton X-100. Then the substrate of elastase MeOSuc-AAPV-pNA (Sigma, M4765) was added to neutrophil lysate and incubated at 37˚C for 45 min in the dark as described [66]. The absorbance was measured with the wavelength set to 400 nm.

## Neutrophil phagocytosis

For neutrophil phagocytosis, *M. pneumoniae* and *Staphylococcus albus* was fluorescently labeled with CFSE (Invitrogen, C34570) as described previously [67]. Neutrophils were seeded in 96-well plates at a density of $5\times10^5$ cells per well in RPMI 1640 complete medium supplemented with 5 mM itaconate for 1 h at 37˚C, and the cells were infected with CFSE-labeled *M. pneumoniae* or *S. albus* at a MOI of 10. The infection was synchronized by centrifuging the plates at 500 g for 2 min. After 1 h infection at 37˚C, 10 µg/ml azithromycin (MCE, HY-17506) was added to each well for killing the extracellular *M. pneumoniae*, and incubated at 37˚C for 30 min as described [27]. The cells were washed with PBS and percentage of phagocytosis was detected by flow cytometry.

## Neutrophil metabolic flux

Neutrophils were seeded in 96-well plates at a density of $5\times10^5$ cells per well in RPMI 1640 complete medium supplemented with 5 mM itaconate for 1 h at 37˚C, and then infected with *M. pneumoniae* at a MOI of 10. After a 2-h incubation at 37˚C, the OCR and ECAR were measured on a SpectraMax iD3 multi-mode microplate reader (Mol. Devices) by using an extracellular oxygen consumption assay kit (Abcam, ab197243) and a glycolysis assay kit (Abcam, ab197244) according to the manufacturer's instructions, respectively.

## Statistical analysis

The statistical significance was assessed by unpaired *t*test and one-away ANOVA as appropriate using GraphPad Prism 8.0. Correlation analysis was conducted using Spearman correlation. Data are shown as the mean with standard error of mean (SEM), and $P < 0.05$ is considered statistically significant.

## Supporting information

**S1 Fig. Deficiency of *Irg1* decreased lung neutrophils in mice during *M. pneumoniae* infection.** The experiment was described as in Fig 1D–1J. (A) Representative flow cytometric analysis of CD11b+Ly6G+ neutrophils gated on CD45+ cells in BALF from WT and *Irg1* KO mice. (B) Neutrophil numbers in BALF (n = 5–8), pooled from three independent experiments. (C) Neutrophil numbers in lung (n = 5–8), pooled from three independent experiments. (D) Alveolar macrophages (AM) numbers in BALF (n = 5–8), (E) AM numbers in lung (n = 4–5), pooled from three independent experiments. Data are presented as mean ± SEM. Statistical significance tested by one-way ANOVA test (*, p<0.05; **, p<0.01; ***, p<0.001).
(TIF)

**S2 Fig. Effect of neutrophil depletion on monocyte/alveolar macrophage numbers and *M. pneumoniae* burden after infection in mice, and *M. pneumoniae* infection upregulates *Irg1*/itaconate pathways in neutrophils *in vitro*.** Flow cytometric analysis of IRG1 expression in lung alveolar macrophages (AM) at day 1 and day 3 after *M. pneumoniae* (MP) infection in mice (n = 4 per group). (B-E) The experiment was described as in Fig 2E–2G (n = 6 per group). (B) Neutrophil numbers in lung. (C) *M. pneumoniae* load in BALF. (D) Alveolar macrophages (AM) numbers in lung. (E) Monocyte numbers in lung. (F and G) Dynamic expression of *Irg1* mRNA expression (F) and IRG1 protein expression (G) in mouse BM neutrophils after *M. pneumoniae* (MOI = 0 or 10) infection *in vitro*. The experiment was performed three times independently. (H) Itaconate concentrations in the culture supernatants of mouse neutrophils after *M. pneumoniae* infection for 20 h. (I) Mouse BM neutrophils were treated with live *M. pneumoniae*, heat (96˚C for 10 minutes)-killed *M. pneumoniae* and 5 μg/ml *M. pneumoniae* lipid-associated membrane proteins (LAMP) for 12 h followed by western blot analysis of the IRG1 protein, representative of three experiments. (J) Itaconate concentrations in the culture supernatants of human neutrophils after *M. pneumoniae* infection for 20 h. Data are presented as mean ± SEM. Statistical significance tested by unpaired, two-tailed Student's *t* test (*, p<0.05; **, p<0.01; ***, p<0.001).
(TIF)

**S3 Fig. Representative flow cytometric analysis of transferred neutrophils in the mouse lung and BALF.** Sorted BM neutrophils (CD45.2 background) from WT and *Irg1* KO mice were intravenously transferred into CD45.1/2 mice, respectively. Then, the recipient mice were i.n. infected by *M. pneumoniae* (MP) or treated with PPLO medium (Mock group). Mice were sacrificed at day 1 after infection to analyze transferred neutrophils (CD45.1-CD45.2+ cells) and non-transferred neutrophils (CD45.1+CD45.2+ cells) in BALF (A) and lung (B) by flow cytometry.
(TIF)

**S4 Fig. Adoptive transfer of *Irg1* KO neutrophils did not affect the body weight of recipient mice during *M. pneumoniae* infection.** The experiment was described as in Fig 3. Data are pooled from two independent experiment and are presented as mean ± SEM.
(TIF)

**S5 Fig. Adoptive transfer of neutrophils from WT mice aggravates *M. pneumoniae* pneumonia in *Irg1* KO mice.** (A-E) Sorted BM neutrophils from WT and *Irg1* KO mice were intravenously transferred into *Irg1* KO mice followed by *M. pneumoniae* (MP) infection. Mice were sacrificed at day 3 after *M. pneumoniae* infection (n = 6 per group). (A) Neutrophil numbers in BALF. (B) Neutrophil numbers in lung. (C) Itaconate concentrations in BALF. (D) *M. pneumoniae* load in BALF. (E) LDH levels in BALF. (F) Representative H&E staining of lung tissue. Scale bar, 50 μm. (G) Histological scores. Data are pooled from two independent experiment and are presented as mean ± SEM. Statistical significance tested by unpaired, two-tailed Student's *t* test (*, p<0.05; **, p<0.01; ***, p<0.001).
(TIF)

**S6 Fig. β-glucan inhibits M. pneumoniae-induced *Irg1* expression in neutrophils *in vitro*.** (A and B) Mouse BM neutrophils were pretreated with β-glucan (0 or 5 μg/ml) for 1 h followed by *M. pneumoniae* (MP) infection for 12 h. The experiment was performed three times independently. (A) The *Irg1* mRNA expression in neutrophils. (B) IRG1 protein expression in neutrophils, representative of three experiments. (C) Itaconate concentrations in the culture supernatants. Data are presented as mean ± SEM. Statistical significance tested by unpaired, two-tailed Student's *t* test (*, p<0.05; **, p<0.01; ***, p<0.001).
(TIF)

**S7 Fig. Itaconate inhibits neutrophils mtROS production, but does not affect NADPH oxidase activity and lipid ROS production upon *M. pneumoniae* infection.** (A) Flow cytometric analysis of the total ROS of lung (alveolar macrophages) AM from WT and *Irg1* KO mice at day 3 after *M. pneumoniae* (MP) infection with or without exogenous itaconate (ITA) treatment (n = 4 or5 per group). Data are presented as mean ± SEM. Statistical significance tested by unpaired, two-tailed Student's *t* test. (B and C) Mouse BM neutrophils from WT and *Irg1* KO mice were pretreated with 5 mM itaconate, 1 μM MitoPQ, or 100 μM MitoTEMPO for 1 h followed by the infection with *M. pneumoniae* for 12 h. Total ROS (B) and mtROS (C) production were determined by flow cytometry. (D) Neutrophil NADPH oxidase activity in neutrophils after 2 h infection with *M. pneumoniae* (MOI = 0 or 10) in the presence of itaconate (0–5 mM), 10 μM NADPH oxidase inhibitor DPI, 1 mM NADPH oxidase activator PMA. (E) Neutrophil lipid ROS production in neutrophils after 2 h infection with *M. pneumoniae* (MOI = 0 or 10) in the presence of 5 mM itaconate, 10 μM ferroptosis inhibitor ferrostatin-1 (Fer-1), 10 μM ferroptosis activator (1S,3R)-RSL3 (RSL3). Data are pooled from three independent experiments and are presented as mean ± SEM. (B-E) Statistical significance tested by one-way ANOVA test (*, p<0.05; **, p<0.01; ***, p<0.001).
(TIF)

**S8 Fig. Itaconate inhibits neutrophil mtROS production without affecting M. *pneumoniae* growth and neutrophil elastase activity.** (A) *M. pneumoniae* (MP) detection after 5 mM itaconate (ITA), 1 μM MitoPQ, 100 μM MitoTEMPO, or 10 μM DPI treatment for 2 h. (B) Mouse BM neutrophils and (C) human neutrophils were pretreated with 5 mM itaconate, 1 μM MitoPQ, 100 μM MitoTEMPO, 10 μM DPI for 1 h followed by *M. pneumoniae* infection for 2 h, and neutrophils mtROS were detected by flow cytometry. (D) Bacterial phagocytosis by neutrophils. *Staphylococcus albus* (*S. albus*) was used as a positive control. (E and F) Neutrophils in the presence of 5 mM itaconate for 1 h followed by *M. pneumoniae* infection for 2 h. (E) cells were stained with anti-CD63 antibodies and detected by flow cytometry for neutrophil degranulation, (F) and neutrophil elastase activity was measured with OD 400 nm. Data are pooled from three independent experiments and are presented as mean ± SEM. Statistical

significance tested by one-way ANOVA test (*, p<0.05; **, p<0.01; ***, p<0.001).
(TIF)

**S9 Fig. MtROS modulates neutrophil apoptosis and itaconate does not affect neutrophil glycolysis.** (A and B) Mouse BM neutrophils were treated with 1 μM MitoPQ or 100 μM Mito-TEMPO *in vitro* for 12 h. Neutrophil mtROS production (A) and the percentage of apoptotic cells (B) were pooled from three independent experiments. Statistical significance tested by one-way ANOVA test. (C) The number of neutrophils in the lung from WT and *Irg1* KO mice at day 1 after *M. pneumoniae* (MP) infection (n = 4–6 per group), pooled from two independent experiments. Statistical significance tested by unpaired, two-tailed Student's *t* test. (D) The mRNA expression of *Cxcl1*, *Cxcl2* and *Csf3* in the mouse neutrophils from WT and *Irg1* KO mice at day 1 after *M. pneumoniae* infection (n = 4–6 per group), pooled from two independent experiments. Statistical significance tested by unpaired, two-tailed Student's *t* test. (E) The mRNA expression of *Il-1β*, *Il-6*, and *Tnf-α* in the mouse neutrophils of WT and *Irg1* KO mice after 12 h of infection with *M. pneumoniae*, pooled from three independent experiments. Data are presented as mean ± SEM. Statistical significance tested by unpaired, two-tailed Student's *t* test. (F) Neutrophil extracellular acidification rate (ECAR) after 2 h infection with *M. pneumoniae* (MOI = 0 or 10) in the presence of itaconate (ITA, 0 or 5 mM), pooled from three independent experiments. RFU: relative fluorescence units. *, p<0.05; **, p<0.01; ***, p<0.001.
(TIF)

**S1 Table. Demographic and clinical information of patients with severe *M. pneumoniae* pneumonia.**
(DOCX)

**S1 Data. Source data for Figs 1A, 1C–1H, 1J, 2A, 2B, 2E, 2G, 2H, 2J, 2K, 3A–3F, 3H, 4A–4H, 4J, 5A–5H, 6A-6E, 7A and 7C.**
(XLSX)

**S2 Data. Source data for S1B–S1E, S2A–S2F, S2H, S2J, S4, S5A–S5E, S5G, S6A, S6C, S7A–S7E, S8A–S8F, S9A-S9F Figs.**
(XLSX)

**S1 File. Flow cytometry gating.**
(PDF)

## Acknowledgments

We thank Prof. Jie Zhou at Tianjin Medical University for providing CD45.1 mice. We thank Zhaoxiang Du, Weixiang Long, Dr. Yi Cao, and Dr. Gaojian Lian for their help in our animal experiments. We thank Prof. Shunli Gao for her help in collecting the clinical samples.

## Author Contributions

**Conceptualization:** Aihua Lei.

**Data curation:** Cui Wang, Aihua Lei.

**Formal analysis:** Cui Wang, Zijun Yan, Yujun Zhou, Aihua Lei.

**Funding acquisition:** Haijun Zhang, Aihua Lei.

**Investigation:** Cui Wang, Jun Wen, Zijun Yan, Yujun Zhou, Zhande Gong, Ying Luo, Zhenkui Li, Aihua Lei.

**Methodology:** Cui Wang, Aihua Lei.

**Resources:** Jun Wen, Kang Zheng, Haijun Zhang, Nan Ding, Chuan Wang, Cuiming Zhu, Yimou Wu, Aihua Lei.

**Supervision:** Aihua Lei.

**Validation:** Cui Wang, Aihua Lei.

**Writing – original draft:** Cui Wang, Aihua Lei.

**Writing – review & editing:** Cui Wang, Jun Wen, Zijun Yan, Yujun Zhou, Zhande Gong, Ying Luo, Zhenkui Li, Kang Zheng, Haijun Zhang, Nan Ding, Chuan Wang, Cuiming Zhu, Yimou Wu, Aihua Lei.

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
