## [Decision Letter · Decision Letter 0]

28 May 2024

Dear Professor Lei,

Thank you very much for submitting your manuscript "Suppressing neutrophil itaconate production attenuates Mycoplasma pneumoniae pneumonia" for consideration at PLOS Pathogens. As with all papers reviewed by the journal, your manuscript was reviewed by members of the editorial board and by several independent reviewers. In light of the reviews (below this email), we would like to invite the resubmission of a significantly-revised version that takes into account the reviewers' comments.

All reviewers found the manuscript to be interesting, well-written, and potentially valuable. However, two of the reviewers identified a number of significant shortcomings in terms of description of important experimental details, and reviewer #3 identified at least two experiments (associated with Figures 3 and 4) that would make the findings a lot clearer. Finally, given that Mycoplasma pneumoniae is a human pathogen but all the findings are in a mouse model, it is important to add some discussion of how these findings relate to both human disease and any other studies involving human neutrophils. Please address all the reviewers' major and minor comments in a revised version of this manuscript.

We cannot make any decision about publication until we have seen the revised manuscript and your response to the reviewers' comments. Your revised manuscript is also likely to be sent to reviewers for further evaluation.

Sincerely,

Mitchell F. Balish, Ph.D.

Academic Editor

PLOS Pathogens

Thomas Guillard

Section Editor

PLOS Pathogens

Michael Malim

Editor-in-Chief

PLOS Pathogens

orcid.org/0000-0002-7699-2064

All reviewers found the manuscript to be interesting, well-written, and potentially valuable. However, two of the reviewers identified a number of significant shortcomings in terms of description of important experimental details, and reviewer #3 identified at least two experiments (associated with Figures 3 and 4) that would make the findings a lot clearer. Finally, given that Mycoplasma pneumoniae is a human pathogen but all the findings are in a mouse model, it is important to add some discussion of how these findings relate to both human disease and any other studies involving human neutrophils. Please address all the reviewers' major and minor comments in a revised version of this manuscript.

Reviewer's Responses to Questions

**Part I - Summary**

Reviewer #1: Wang and colleagues report on the role of ACOD1/Irg1 and itaconate in a mouse model of Mycoplasma pneumoniae infection. While itaconate production by macrophages has been in the focus of several studies on bacterial infection in the mouse model, this study focusses on Acod1 expression and itaconate production by neutrophils. Further, while itaconate production has been shown to be protective in other bacterial infection models due to its antimicrobial and/or immunoregulatory activity, in the M. pneumoniae model reported here, the opposite effect is observed with reduced microbial burden, cytokine production and tissue damage in Acod1-/- mice. These results are novel and interesting, as they suggest that itaconate has proinflammatory activity in certain infections. The authors provide evidence that interventions to reduce itaconate production during infection (e.g. transfer of Acod1-/- neutrophils, treatment with b-glucan) have a beneficial effect in the mouse infection model. They also show that itaconate inhibits mitochondrial ROS production by neutrophils and link this to reduced killing of M. pneumoniae. The study is well designed and experiments are well performed, the manuscript is written very well and the sequence of results are easy to follow. There are number of issues concerning the description and interpretation of the results that need to be addressed.

Reviewer #2: This manuscript describes a series of studies that demonstrate that itaconate-producing neutrophil promotes Mycoplasma pneumoniae pneumonia in a mouse model. Overall this is a well written manuscript that clearly states the rationale for each of the studies and their results.

The strength of this manuscript is that the authors have used multiple approaches to corroborate their results, i.e., it is not one experiment but multiple approaches to confirm the results. Thus the data appear strong.

Another strength is the overall conclusions are well supported and significant. It is possible that if proven true in human M. pneumoniae infection, then this could be a possible target for therapy. It may also be applicable to other diseases where excessive inflammation may contribute to pathology.

One limitation is that these studies were done in a murine model of M. pneumoniae disease, which is not a great model for the human disease. However, it is one of the best available. So, a statement about this limitation and how results may or may not lead to understanding in human disease would be interesting and I think strengthen the manuscript.

Reviewer #3: The paper "Suppressing neutrophil itaconate production attenuates Mycoplasma pneumoniae pneumonia" by Wang et al investigates the role of neutrophils and their production of itaconite during Mycoplasma pneumoniae pneumonia (MPP).

A major strength of the study is the use of different in-vitro and in-vivo mice models and assays (such as Irg1- KO mice, pharmacological inhibitors etc) next to the use of human and MPP-patient derived neutrophils to interrogate the source and role of itaconate during MPP.

Weaknesses: poor description of patient cohort in terms of disease severity; days after onset of symptoms is not clear. Discussion could benefit from context: what is know about itaconate-producing neutrophils in airway diseases/infections? what about other myeloid cells: do they contribute to the process?

also considering 1) the findings in patients with severe MPP (Fig2J);

and 2) the experiments with the Dectin1-ligand beta-glucan: are observed effects on Irg1 expression due to direct effects of beta-glucan on neutrophils or indirect via monocytes/macrophages?

**Part II – Major Issues: Key Experiments Required for Acceptance**

Reviewer #1: Specific comments:

Figure 1: mice are infected with 10e8 CFU, and when measured 3 days later, the BAL fluid contains less than 10e6 CFU/ml (Fig. 1D). Therefore, it appears that only a quite small fraction of the infecting bacteria is left after three days of infection. How many bacteria are recovered from the lung tissue at this time? Which volume was used to obtain the BALF cells?

The histopathology (Fig. 1G) shows increased leukocyte infiltration in the WT mice and after administration of ITA in representative images. To be informative, quantitative data from analysis of several sections from all the mice in the experiment are required. These could be obtained e.g. by measuring the area of air-filled alveoli vs. cellular infiltrates.

The intranasal application of 10 mg/kg itaconate increases CFU and cytokine responses in Acod1-/- mice. It would be informative to also show itaconate concentrations in the BALF of treated mice in Fig. 1C.

Figure 2: the data convincingly demonstrate that itaconate production in the lungs of M. pneumoniae-infected mice is primarily derived from neutrophils. For human patients with severe pneumonia, expression of IRG1 protein in neutrophils is also shown by Western blot and intracellular staining. It is remarkable that the concentration of itaconate measured in the BALF of human patient is much lower than in mice (20 nM vs. 1000 nM). This apparent lower capacity of human ACOD1 to produce itaconate would be in line with earlier reports for human macrophages (Michelucci 2013 PNAS, Kohl 2023 EMBO Mol Med) and could be due to the reduced enzymatic activity reported previously (Chen…Pessler 2019 PNAS). This difference between species should be discussed. In addition, the content of itaconate in lysates and culture supernatants of mouse vs. human neutrophils should be analyzed to validate production of itaconate in humans in comparison with the murine system.

Figure 3: the approach to use adoptive transfer of wt and Acod1-/- neutrophils to corroborate the function of neutrophil-derived itaconate is valid. The results are consistent with the hypothesis, yet it is not demonstrated that the injected cells are indeed traveling to the lung or cause the effect on M. pneumoniae by a different mechanism. Therefore, the presence of transferred neutrophils in lung and BALF should be confirmed. This can be done using syngeneic markers like CD45.1/2. In addition, it should be tested whether itaconate in BLAF is indeed higher after transfer of wt neutrophils.

For the histopathology in Fig. 3F, the same criticism as in Fig. 1 applies. Please, provide quantitative data allowing statistical analysis.

Figure 4: the treatment of infected mice with b-glucan has beneficial effects that are associated with reduced expression of Irg1 in BALF neutrophils and itaconate levels in the BALF. Whether this association reflects a causal link remains unclear because b-glucan can have many and diverse effects on the response to bacterial infection. E.g. the number of neutrophils recruited to the lung is very strongly reduced by b-glucan treatment which may be the major contributor to the effect on bacterial burden. To support the claim that b-glucan acts mainly as a IRG1 inhibitor, a direct reduction of IRG1 expression and itaconate production by neutrophils treated win vitro with b-glucan should be tested.

Reviewer #2: None

Reviewer #3: 1. The patient cohort is poorly described. Basic characteristics are shown in Suppl Table 1, but nothing is mentioned about how severe disease is diagnosed:what about chest radiograph (CXR) findings and related CXR severity scores, hypoxemia requiring oxygen supply, and inflammatory parameters? Also it is unclear at what stage after onset of symptoms was BALF sampled for itaconate concentrations vs # neutrophils (Fig2J)? Also, findings shown in Fig 2J should be better discussed.

2. Adoptive transfer of Igr1-KO neutrophils attenuates MPP: read out is done on day 3 p.i.

-Are effects still effective at later stages in the disease?

-is attenuated MPP also reflected in less weight loss in mice?

-how many neutrophils were transferred?

Would have expected a few neutrophils in lungs from control mice, especially after iv injection.

3. Did the authors not expect that suppletion with exogenous ITA would result in more severe inflammation/pathology in the lungs (Fig 1)?

**Part III – Minor Issues: Editorial and Data Presentation Modifications**

Reviewer #1: Figure 6: the header for this part in the Results section reads “Itaconate inhibits neutrophil survival” which is the opposite of what is shown in the figure.

Figure 7: the use of knockout neutrophils instead of the chemical inhibitors for TLR2/4 and Myd88. It is not stated whether the phosphorylation of STAT1 refers to tyrosine or serine.

Reviewer #2: There are some minor weaknesses/revisions needed:

1) Line 256, heading: States "Itaconate inhibits neutrophil survival during M. pneumoniae infection", but the data shows that itaconate inhibits apoptosis. Thus, the header needs to be revised.

2) Make sure that all abbreviations in figures are defined in figure legends. For example, Figure 2D has the abbreviation "Lung leukocytes-Neu". I initially interpreted this as designating only lung neutrophils, but it actually refers to lung leukocytes depleted of neutrophils. This should be clarified in the figure legend. Please review the other figures.

3) Format of references need to be consistent. There are several references where every first letter of word is capitalized, while others do not.

4) Need to italicize all species names in references, e.g. Mycoplasma pneumoniae, Staphylococcus aureus, etc.

Reviewer #3: 1. Fig 5C: how is % SDH activity calculated for each group? I would have expected that MP infection would induce SDH activation and thus result in higher levels compared to DMSO-PBS group

2. lines 107-112: describe what tissue is analysed

Line 115: typo: "with intranasally administrated with "

Line 116: lower level of MP load > should be "lower load of MP bacteria"

line 117: what is a "pure MP infection"?

line 392: "regents" > reagents

PLOS authors have the option to publish the peer review history of their article (what does this mean?). If published, this will include your full peer review and any attached files.

Reviewer #1: **Yes: **Roland Lang

Reviewer #2: **Yes: **Jerry W. Simecka, Ph.D.

Reviewer #3: No
---

## [Decision Letter · Decision Letter 1]

22 Aug 2024

Dear Professor Lei,

Thank you very much for submitting your manuscript "Suppressing neutrophil itaconate production attenuates Mycoplasma pneumoniae pneumonia" for consideration at PLOS Pathogens. As with all papers reviewed by the journal, your manuscript was reviewed by members of the editorial board and by several independent reviewers. The reviewers appreciated the attention to an important topic. Based on the reviews, we are likely to accept this manuscript for publication, providing that you modify the manuscript according to the review recommendations.

The revised version of the paper is much improved, but reviewer #3 has some small points to address. In particular, the first point requires a small amount of explanation, and the third warrants some clarification. Once these minor issues are satisfactorily addressed, I see no barrier to publication.

Sincerely,

Mitchell F. Balish, Ph.D.

Academic Editor

PLOS Pathogens

Thomas Guillard

Section Editor

PLOS Pathogens

Michael Malim

Editor-in-Chief

PLOS Pathogens

orcid.org/0000-0002-7699-2064

The revised version of the paper is much improved, but reviewer #3 has some small points to address. In particular, the first point requires a small amount of explanation, and the third warrants some clarification. Once these minor issues are satisfactorily addressed, I see no barrier to publication.

Reviewer Comments (if any, and for reference):

Reviewer's Responses to Questions

**Part I - Summary**

Reviewer #1: The authors have addressed all my comments and provided additional data that support their conclusions.

They even allowed me to contribute to the Discussion section by copy/pasting my comment on mouse/human differences related to Figure 2. Thus, I am more than satisfied.

Reviewer #2: The authors appear to have address the concerns raised by the reviewers. They have addressed my comments well. It is a very interesting paper which may have identified a mechanism used by M. pneumoniae to persist along the respiratory tract, despite an intense inflammatory response.

Reviewer #3: The authors have adequately answered most of the questions; unfortunately a few items yet remain to be better explained:

1. Fig S3 showing flowcytometry plots of neutrophils in lung and BALF cells to indicate the presence of transferred donor neutrophils in these tissues. Hereto the authors used CD45 congenic mice.

Why were CD45.1/.2 chosen as recipient? how can the transferred CD45.1 donor cells be distinguished? Ideally CD45.1 are transferred into CD45.2 recipients, leaving two single positive populations in the CD45.1 vs CD45.2 plots.

What antibodies were used to stain for CD45.1 and CD45.2 (not present in methods).

2. The answer to whether itaconite may be effective at later stages in infection is not correct in my opinion: the authors cite studies demonstrating effects on CD8 T cell responses. however, ITA affects the priming of T cells (so in early stages) leaving "crappy CD8 effector T cells" that cannot resolve the infection at the later stages. So you could argue it's a matter of semantics, as the crappy effector T cells are an effect of ITA. however in the Discussion, lines 398-400, the authors phrased it better.

3. Regarding the suppletion of exogenous ITA in wt mice and tissue damage: apparently the effects of ITA are not dose-dependent; its presence as such may be enough to evoke damage, or there may be a certain threshold? this should be better worded in the Discussion than is done now: lines 381-384: The fact that supplementation with exogenous itaconate did not lead to more severe lung inflammation in WT mice suggests that the endogenous itaconate production induced by M. pneumoniae may be sufficient to cause the effects.

Minor:

-MFI=mean fluorescenCE intensity, not fluorescent.

-in methods "Mycoplasma pneumoniae and infection: add "mouse model"

**Part II – Major Issues: Key Experiments Required for Acceptance**

Reviewer #1: (No Response)

Reviewer #2: None

Reviewer #3: (No Response)

**Part III – Minor Issues: Editorial and Data Presentation Modifications**

Reviewer #1: (No Response)

Reviewer #2: No major concerns.

Reviewer #3: (No Response)

PLOS authors have the option to publish the peer review history of their article (what does this mean?). If published, this will include your full peer review and any attached files.

Reviewer #1: No

Reviewer #2: **Yes: **Jerry W. Simecka

Reviewer #3: **Yes: **WWJ Unger

Figure Files:

Data Requirements:

Reproducibility:

References:

---

## [Editor Report · Decision Letter 2]

24 Sep 2024

Dear Professor Lei,

We are pleased to inform you that your manuscript 'Suppressing neutrophil itaconate production attenuates Mycoplasma pneumoniae pneumonia' has been provisionally accepted for publication in PLOS Pathogens.

Best regards,

Mitchell F. Balish, Ph.D.

Academic Editor

PLOS Pathogens

Thomas Guillard

Section Editor

PLOS Pathogens

Michael Malim

Editor-in-Chief

PLOS Pathogens

orcid.org/0000-0002-7699-2064
---

## [Editor Report · Acceptance letter]

2 Oct 2024

Dear Professor Lei,

We are delighted to inform you that your manuscript, "Suppressing neutrophil itaconate production attenuates Mycoplasma pneumoniae pneumonia," has been formally accepted for publication in PLOS Pathogens.

Best regards,

Michael Malim

Editor-in-Chief

PLOS Pathogens

orcid.org/0000-0002-7699-2064